# A computational approach to evaluate how molecular mechanisms impact large-scale brain activity

Maria Sacha[1,3], Federico Tesler[1,3], Rodrigo Cofre [1,2] & Alain Destexhe [1] ✉

Assessing the impact of pharmaceutical compounds on brain activity is a critical issue in contemporary neuroscience. Currently, no systematic approach exists for evaluating these effects in whole-brain models, which typically focus on macroscopic phenomena, while pharmaceutical interventions operate at the molecular scale. Here we address this issue by presenting a computational approach for brain simulations using biophysically grounded mean-field models that integrate membrane conductances and synaptic receptors, showcased in the example of anesthesia. We show that anesthetics targeting $GABA_A$ and NMDA receptors can switch brain activity to generalized slow-wave patterns, as observed experimentally in deep anesthesia. To validate our models, we demonstrate that these slow-wave states exhibit reduced responsiveness to external stimuli and functional connectivity constrained by anatomical connectivity, mirroring experimental findings in anesthetized states across species. Our approach, founded on mean-field models that incorporate molecular realism, provides a robust framework for understanding how molecular-level drug actions impact whole-brain dynamics.

Recent advances in neuroimaging techniques have made it possible to build detailed brain atlases and generate personalized connectomes of the human brain through cutting-edge tools such as diffusion magnetic resonance imaging (MRI), magnetoencephalography and functional MRI (fMRI). These advancements have catalyzed the conceptualization within the scientific community of highly detailed, personalized whole-brain models, colloquially termed Digital Twins, poised to revolutionize the exploration, diagnosis and therapeutic interventions for various neurological conditions, including disorders of consciousness, epilepsy, Alzheimer's disease and schizophrenia[1–5].

Despite substantial advances in this domain, a primary challenge faced by these models lies in effectively addressing the diverse spatial and temporal scales inherent to brain activity. Numerous diseases or alterations in brain states emerge at the molecular or cellular level,

requiring comprehensive descriptions within whole-brain models to ensure accuracy. Thus, integrating cellular-scale details is imperative for these models to capture the intricacies of neurological phenomena. However, the human brain comprises approximately 85 billion neurons and an estimated $10^{15}$ synaptic connections. Consequently, creating a large-scale model of the brain (macroscale) with single-cell resolution (microscale) remains an insurmountable challenge. Establishing a seamless interface between these disparate scales is, therefore, a pivotal challenge for enhancing the utility and precision of digital twins and whole-brain simulations.

This Article introduces a framework aimed at addressing this challenge, offering an efficient pathway for transitioning from the cellular scale to the whole-brain level. Central to our methodology is the integration of a biophysically grounded mean-field model,

[1]Paris-Saclay University, CNRS, Paris-Saclay Institute of Neuroscience (NeuroPSI), Saclay, France. [2]Present address: Université Côte d'Azur, INRIA CRONOS Team, Sophia Antipolis, France. [3]These authors contributed equally: Maria Sacha, Federico Tesler. ✉e-mail: alain.destexhe@cnrs.fr

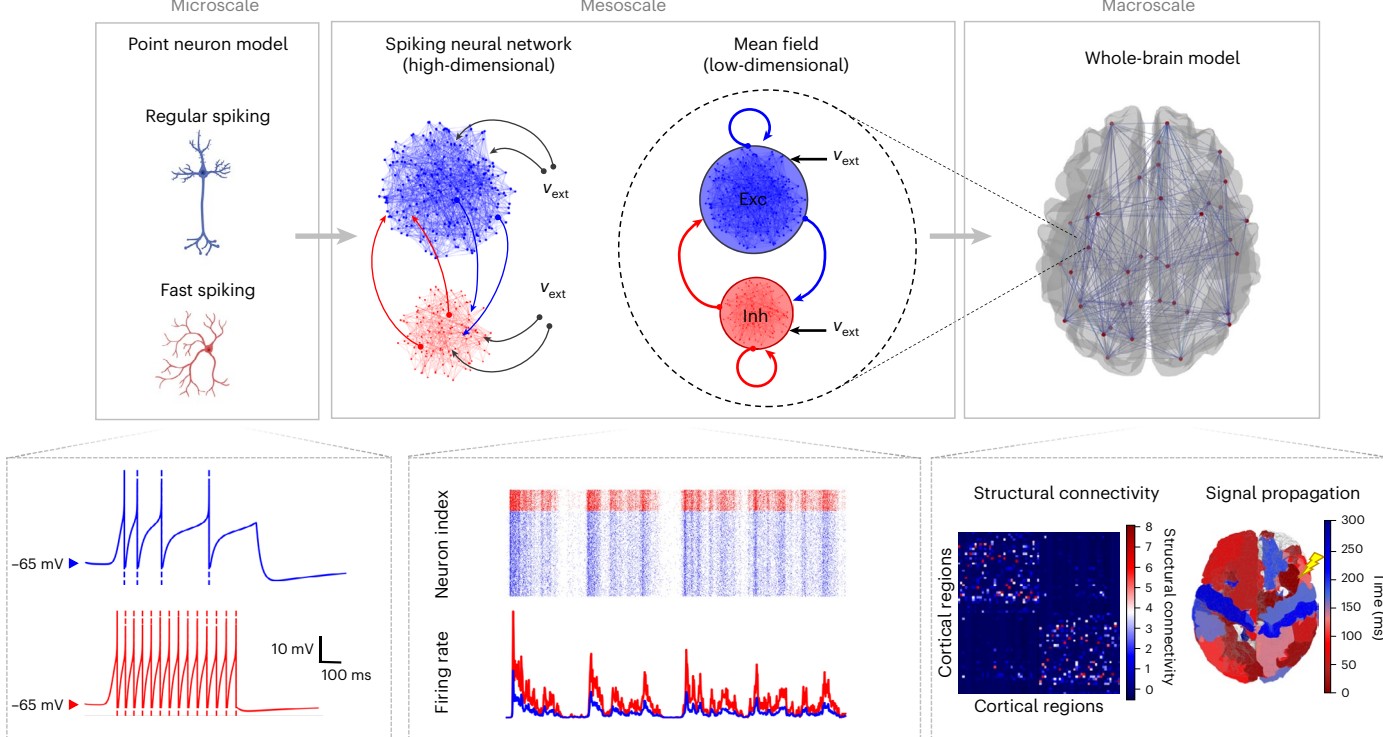

**Fig. 1 | Schematic diagram of the bottom-up construction of the whole-brain modeling framework (top row) and the corresponding signals simulated across the various scales (bottom row).** Microscale: a point-neuron model is selected to describe the activity of individual neurons, representing different cellular types with specific neuroreceptors and characterized by distinct firing patterns—ranging from RS cells (Exc, excitatory, blue) with adaptive firing rates to FS cells (Inh, inhibitory, red). Mesoscale: based on experimental data of brain microcircuits, we connect the single neurons to build spiking neural networks (high-dimensional mesoscale) from which biologically grounded mean-field models are derived to describe collective dynamics (low-dimensional mesoscale); here $v_{ext}$ represents an external input. Macroscale: SC data are integrated into the framework to build large-scale networks of mean-field models to perform whole-brain simulations, allowing the exploration of emergent properties at this scale (for instance, responsiveness to external stimulation). Figure created with BioRender.com.

encompassing cellular and molecular properties. This model effectively reduces system dimensionality while capturing emergent properties observed at the meso- and macroscale due to cellular-level changes. Our methodology comprises four parts: (1) single-cell models (microscale); (2) networks of spiking neurons (mesoscale); (3) a mean-field model elucidating the activity and dynamics of the network (mesoscale, low-dimensional); and (4) a whole-brain model leveraging the mean-field model for generating large-scale simulations (macroscale). To facilitate the execution of these simulations, we utilize The Virtual Brain (TVB) platform[6], a versatile tool enabling the execution of large-scale simulations with customizable brain connectomes.

To illustrate our approach, we simulate the molecular actions leading to general anesthesia, specifically with anesthetics such as ketamine and propofol, and investigate their effects at the mesoscale and whole-brain levels. The action of these anesthetics is incorporated here by modeling their effects on membrane receptors at the single-cell level (a detailed simulation of the molecular action of drugs could also be incorporated but is not adopted for the presentation of the framework in this Article). Our focus includes the generation of slow-wave activity, reduced responsiveness to external stimuli and increased anatomical–functional correlations, as measured experimentally for anesthetized states.

## Results

The components of our method are introduced sequentially, following the bottom-up approach of the model construction. As a benchmark, we apply this model within the context of general anesthesia, which serves as an example of how molecular changes at the synaptic level

can substantially impact whole-brain dynamics. The diagram of the framework is shown in Fig. 1.

### Framework overview

**Single-cell model.** The first step of our framework consists of selecting a spiking neuron model. Here, we used the Adaptive Exponential integrate-and-fire (AdEx) model, recognized for its capacity to capture a wide range of neuronal firing patterns, while also providing biophysically realistic representations of neuronal behavior and mathematical simplicity[7,8]. Nevertheless, our methodology remains adaptable to a variety of neuronal models, such as quadratic Integrate-and-Fire or Hodgkin–Huxley[9–11].

The AdEx model consists of two differential equations. The first equation (equation (1)) describes the membrane potential dynamics and the second (equation (2)) a slow adaptation current (see Methods for details). This adaptation current varies over time in response to spiking activity, modulating the excitability of the neuron[7].

We consider two types of cell: pyramidal (excitatory) neurons and fast-spiking (FS, inhibitory) interneurons. Each type of cell is characterized by a different set of parameters in the AdEx model. The baseline parameters used for each neuron type are extracted from fits on experimental data of cortical neurons[12]. However, the method we present can be equally applied to simulate the dynamics of neurons in other brain areas such as the thalamus[13], basal ganglia[14] or cerebellum[11].

**Network of spiking neurons.** The second step of our framework consists of the construction of networks of spiking neurons representing local circuits or specific brain structures. The networks are characterized by the type and number of neurons, their connectivity patterns (probability

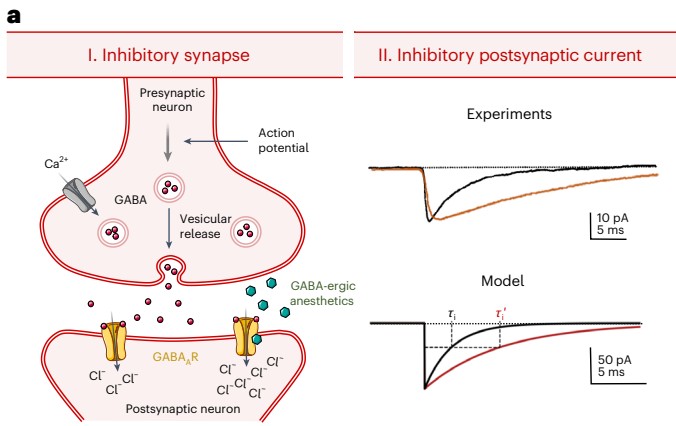

**Fig. 2 | Synaptic targets of general anesthesia and its effect at the cellular scale, in experiments and modeling. a**, General anesthesia that targets GABA$_A$ receptors (for instance, propofol) prolongs IPSPs by increasing the opening duration of the chloride channels. We model this effect as an increase in the inhibitory synaptic decay time constant ($\tau_i$). We illustrate the comparison of the model with whole-cell patch clamp recordings demonstrating the prolongation of IPSP caused by halothane. **b**, NMDA blockers (for instance, ketamine and xenon) decrease the duration of excitatory postsynaptic potentials (EPSPs), as shown by patch-clamp recordings of AMPA and NMDA currents under the effect of ketamine. This effect can be modeled as a decrease of the excitatory synaptic decay time constant ($\tau_e$). Panels adapted with permission from: **a**(II) (top), ref. 61, Wiley; **b**(II) (top), ref. 62, MIT Press.

of connections between neurons) and the synaptic model used. In terms of the number of neurons and the connectivity pattern, the example presented in this Article adopts values representative of prototypical cortical circuits, specifically modeling a single cortical column. We consider a network made of 10,000 neurons, with 80% of pyramidal cells and 20% of interneurons, connected randomly with probability $P = 0.05$ (ref. 15). Different connectivity patterns can be considered and are compatible with our method, which can be used to represent different brain structures[11,13,16,17]. Regarding the synaptic model, we consider conductance-based synapses through glutamate α-amino-3-hydroxy-5-methyl-4-isoxazolepropionic acid (AMPA)−$n$-methyl-$D$-aspartate (NMDA) and γ-aminobutyric type A (GABA$_A$) receptors, but current-based synapses are also compatible with our framework[18]. The synaptic model is presented in equation (4) in Methods.

**Mean-field model.** The central component of our framework is the integration of a recently developed mean-field formalism, which enables an efficient transition between scales while preserving the characteristics and relevant parameters of both the cellular and network levels. The derivation of the mean-field model used here follows a bottom-up approach based on a second-order Master Equation formalism that describes the average activity (mean firing rate, membrane potential and conductance) and covariance of networks of spiking neurons[18]. A key component of these equations is the transfer function (TF), which relates a neuron's output (firing rate) to its presynaptic input. This function cannot be expressed analytically for complex neurons, particularly those interacting through conductance-based synapses. However, a semi-analytic approach can be followed. First, numerical simulations of the desired neuron model are realized and used to fit an analytically derived mathematical template of the TF. This approach has been applied previously to the AdEx model[12,19], but it has been extended to more complex models, such as the quadratic integrate-and-fire neuron[10] as well as the Morris−Lecar and Hodgkin−Huxley model[9].

Through this semi-analytical approach, population models that preserve biological relevance can be constructed. This formulation allows us to study the effects of changes in molecular and network parameters at the mesoscale level, trace the mechanisms behind dynamical transitions in large populations (facilitated by the low dimension of the mean-field model) and construct whole-brain simulations that incorporate the details and specificities of the micro- and mesoscale at a low computational cost (Supplementary Table 2).

**Whole-brain model.** We move to the last scale of our formalism, meaning the whole-brain scale, which is essential for examining properties that manifest exclusively at a macroscale. The whole-brain model is described by a network of mean-fields interconnected based on experimental data (connectome). First, a parcellation of the brain is selected that divides the cortex into distinct regions. The granularity of the parcellation can vary from tens to thousands of regions. Information on the connection between these regions is available through various anatomical connectivity studies, obtained using diffusion weight imaging, tracer studies (in nonhuman species) or a combination of both. The output of these studies relevant to our method is a connectivity matrix that provides information on the connection strength and the distance between each pair of regions.

The activity of each region can be described using the developed mean-field model, forming a node in the whole-brain model. Each node's input consists of an external noise, described by an Ornstein−Uhlenbeck process, and the output firing rate of its connected nodes. The input of the connected nodes is weighted according to the connection strength and received with a delay based on the distance between the two nodes.

We used a parcellation scheme of a human brain comprising 68 brain regions (Anatomical Desikan−Killiany Atlas[20]), giving rise to a network of interconnected mean fields. The implementation of the model was realized in the TVB platform.

**The case of general anesthesia**

To provide an example of the use of our framework, we consider the case of general anesthesia. General anesthesia directly impact various cortical receptors[21], primarily GABA$_A$ and NMDA receptors[22], influencing the activity of both inhibitory and excitatory synapses. GABAergic anesthetics (propofol and volatile anesthetics) act as a positive allosteric modulator of the GABA$_A$ receptor, potentiating GABA-mediated inhibition[23]. They increase the opening duration of the chloride channel, increasing the Cl$^-$ conductance and prolonging the duration of inhibitory postsynaptic potentials (IPSPs)[24–26]. Conversely, NMDA receptor antagonism (ketamine and xenon) leads to a shortening of the kinetics of the conductance of AMPA−NMDA excitatory synaptic input[27–29]. Thus, the main effect of these anesthetics on cortical synapses can be described as an increase in the time constant of inhibitory synaptic decay ($\tau_i$ in equation (4)) to account for the IPSP prolongation caused by GABAergic anesthetics, while a decrease in

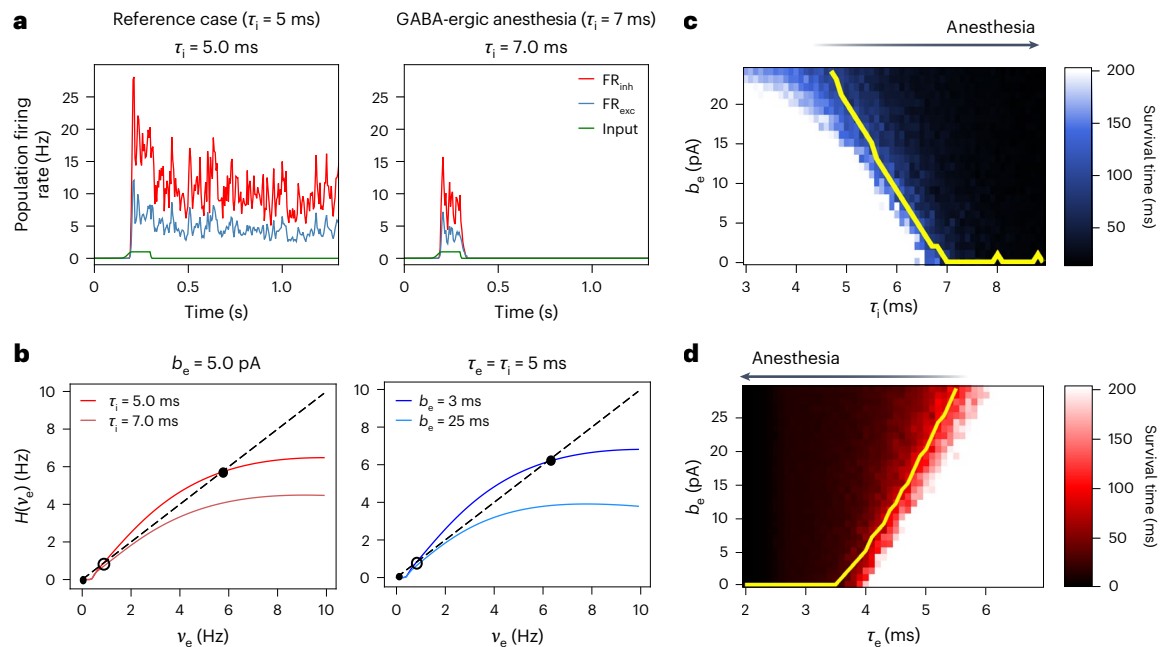

**Fig. 3 | Emergence of UP and DOWN states in simulated anesthesia and NREM sleep: effects of molecular mechanisms at the mesoscale level. a,** Loss of self-sustain activity states in the spiking network as a consequence of changes in GABA-ergic receptors under general anesthesia (for GABA-targeting case). An external input (depicted in green) stimulates the network, resulting in a high firing rate (FR). In the case of higher $\tau_i$ (general anesthesia state, right), the system's activity is suppressed after the end of the stimulus, in contrast to the reference case (lower values of $\tau_i$, left) where a self-sustained state is obtained. **b,** Graphical solution of the first-order mean-field equations for stationary points, predicting the destabilization of self-sustain states and the emergence of the UP and DOWN states in NREM sleep and general anesthesia. The function $H(\nu_e)$ corresponds to the TF of pyramidal cells evaluated at the corresponding stationary solution for FS cells, and the dashed black line corresponds to the bisector (see Methods and ref. 12 for details). Stable fixed points are marked with filled circles and unstable ones with empty circles. The destabilization of the high-rate (self-sustained) stationary point is promoted by the increase of the synaptic decay time $\tau_i$ (left) and due to the increase of spike-triggered adaptation ($b_e$, right). **c,d,** Decay time of the spiking network activity after the end of an external stimulus (survival time of the spiking network) as a function of synaptic decay times $\tau_i$ (**c**) and $\tau_e$ (**d**), and spike-triggered adaptation parameters. For each value of $\tau_{e,i}$, a critical value of $b_e$ exists ($b_{crit}$) for which the sustained network activity is suppressed. For each combination of $b_e$ and $\tau_{e,i}$, 20 different trials with different noise realizations were conducted. The final survival time is reported as the average across these trials. The yellow lines describe the superimposed mean-field predictions of $b_{crit}$ (see text for details), demonstrating the capacity of the mean field to explain the transition in dynamics at the mesoscale level.

the excitatory synaptic decay time ($\tau_e$ in equation (4)) corresponds to NMDA antagonism (Fig. 2).

Despite these seemingly small changes at the cellular scale, at the meso- and macroscale, general anesthesia is characterized by a qualitative transition in the global neuronal dynamics. While the awake state is characterized by asynchronous and irregular dynamics, one of the hallmarks of general-anesthesia states is the appearance of high-amplitude, low-frequency (<4 Hz) oscillations in neuronal activity, correlated with synchronized transitions of neocortical neurons between hyperpolarized and depolarized states[30–32]. These transitions manifest as alternating periods of intense neuronal firing (UP states) and periods of quiescence (DOWN states).

This type of activity is also found in non-rapid eye movement (NREM) sleep[30,33–35], where the low levels of acetylcholine lead to the activation of leak K⁺ channels and the increase of neuronal spike-frequency adaptation (described by the parameter $b$ in the AdEx model), as shown in previous modeling works[12,36]. Variations in the level of acetylcholine are also observed for anesthetics such as propofol[37], but it does not constitute its main pharmacological effect. Thus, how this global transition in neuronal dynamics occurs during anesthesia and how it is related to the molecular changes described before is still an open question. Furthermore, how the molecular changes occurring during NREM sleep lead to the same kind of global transition as in anesthesia is yet not well understood. We show in the next sections how our framework can capture this transition at multiple scales and how the mechanisms and interplay between the different molecular mechanisms acting during anesthesia and sleep can be understood using our framework.

## Emergence of UP and DOWN states in anesthesia and NREM sleep

In this section, we present an analysis of the mechanisms behind the emergence of UP and DOWN states in general anesthesia and its relation to slow waves in NREM sleep. Previous modeling work has shown that the emergence of UP and DOWN states during NREM sleep can be associated with the periodic destabilization of a self-sustained state due to the action of a slow adaptation current[12,38]. For a large-enough strength of the spike-triggered adaptation current, determined by the parameter $b$ in equation (2), a transition from an asynchronous irregular state to a UP and DOWN state is obtained. Here, we extend this analysis and show by combining the mean-field and network simulations (mesoscale level) that this destabilization is promoted by the variations in the synaptic decay times (induced by general anesthesia) and, furthermore, that the transition to UP and DOWN states in general anesthesia can be explained as a result of an interplay between the values of synaptic decay $\tau_{e,i}$ and the spike frequency adaptation of excitatory cells $b_e$ (related to neuromodulatory currents that affect neuronal excitability).

We start by examining the self-sustainability of the AdEx spiking networks as a function of the parameters $b_e$ and $\tau_{e,i}$ ($\tau_e$ or $\tau_i$). To investigate the existence of self-sustained states in the network, we apply a short initial stimulus of 1 Hz of amplitude and a total duration of 120 ms. In the cases where the system exhibits a self-sustained state, the network maintains its activity after the end of the stimulus, otherwise, it decays rapidly to zero (Fig. 3a). An extensive scan along these variables was performed, and the survival time of the activity after the end of the stimulus was calculated. Two parameter scans were performed, one for varied $b_e$ and $\tau_i$ (Fig. 3c) and one for varied $b_e$ and $\tau_e$ (Fig. 3d).

As seen in the Fig. 3c,d, for both cases there is a region of the parameter space where the network sustains its activity beyond the removal of the external input (white area), followed by a transition zone where the survival time decreases progressively until it reaches an area where the network is no longer able to sustain its activity (in black). It becomes evident that the loss of the stable high state of the system depends on the relation between the synaptic parameters $\tau_{e,i}$ and the adaptation $b_e$.

To better understand the interplay of the different parameters ($b_e$, $\tau_e$ and $\tau_i$), and the mechanism behind the emergence of UP and DOWN states, we turn now to the mean-field model (equations (5)–(17)). To this end, an analysis of the solutions (fixed points) of the system is performed. We show in Fig. 3b a graphical solution for the calculation of the fixed points of the excitatory firing rate ($\nu_e$). To obtain this solution, a scan is performed for values of $\nu_e$ and the corresponding function $H(\nu_e) = F_e(\nu_e, \bar{\nu}_i)$ is calculated, where $\bar{\nu}_i = F_i(\nu_e, \bar{\nu}_i)$ (first-order mean field $\dot{\nu}_i = 0$). It is assumed that the adaptation current is at equilibrium, with $W = b\nu_e\tau_w$ (see equation (7)). The fixed point of the system is obtained when $H(\nu_e) = \nu_e$; therefore, the fixed points correspond to the intersection of the plotted function $H$ and the bisector. We see in Fig. 3b (left) that, for $\tau_i = 5$ ms and $b_e = 5$ pA, the system exhibit three intersections corresponding to three fixed points (two stable and one unstable as can be verified). One of the stable solutions is linked to a state of quiescence (no activity), while the second one is linked to a self-sustained state (for all this analysis, it is assumed that no external input is acting on the system). When the value of $\tau_i$ is increased ($\tau_i = 7$ ms in the figure) the system transitioned to a state with a single intersection (single solution), corresponding to a quiescent state; thus the self-sustained solution has disappeared. Similarly, in Fig. 3b (right), the same transition can be observed by changing the value of the constant $b_e$ (for fixed $\tau_{i,e}$). For the cases where the transition occurs, the slow temporal evolution of the adaptation current leads to a periodic emergence and suppression of the self-sustained state (as the system moves back and forth between a three-solution state and a single-solution state), which generates the UP-and-DOWN oscillatory pattern.

In summary, as the magnitude of $b_e$ increases, there is a critical value ($b_{crit}$) for which the system transitions from three fixed points to only one fixed point located at zero. However, this critical value depends on the respective value of $\tau_{e,i}$: variations in $\tau_{e,i}$ modify the global excitability of the system, facilitating or hindering the suppression of the self-sustained state due to $b_e$ (or even completely eliminating the self-sustained state independently of the adaptation). We present the predicted $b_{crit}$ as a function of $\tau_i$ and $\tau_e$ in Fig. 3c,d, respectively, superimposed to the predictions of the survival time of the spiking networks (Fig. 3c,d, yellow lines). As expected, the $b_{crit}$ is lower for higher values of $\tau_i$, with the opposite behavior observed in the case of $\tau_e$. A good agreement is observed between the predictions of the mean field and the spiking network, with the mean-field predictions delimiting the area after which the values of the survival time are uniformly zero. Notice that for spiking networks the intrinsic fluctuations due to limit-size effects induce a gradual transition in the survival time that is not captured by the mean-field model.

In Fig. 4, we demonstrate the transition of states (from asynchronous irregular to UP-and-DOWN dynamics) at the different scales obtained within our framework.

### Emergent whole-brain effects of microscopic changes
We advance to the next stage by using our framework to assess emergent changes at the whole-brain scale. We analyze changes occurring at the functional and behavioral level between the awake and anesthetized states and compare our model with experimental data of both resting state and stimulus–response paradigms.

### State-dependent resting state dynamics—correlation between structural and functional connectivity.
An increase of similarity between the structural and functional connectivity (SC and FC, respectively) during anesthesia has been reported in experimental data[39] and is considered one of the neural signatures of unconsciousness. This property can only be studied using a whole-brain model capable of describing the activity of distinct brain regions (functional component) while incorporating tractographic information on their connections (structural component). Our approach offers the possibility of studying such emergent properties while allowing the mapping of macroscale observations to microscopic factors.

We used resting state fMRI experimental data from humans in the awake state and under propofol anesthesia to reveal differences in SC–FC correlation between the two states. We followed the same analysis on simulated data, by generating blood-oxygenation-level–dependent (BOLD) signals based on the nodes' firing rates of the whole-brain model, and compared the experimental results with the model's predictions.

Figure 5 shows the Pearson correlation between FC and structural SC for both experimental and simulated data in humans. Resting-state fMRI data from 26 individuals during wakefulness and under propofol anesthesia were analyzed. From these signals, the FC matrix was computed for each participant and correlated with a reference SC that comes from a cohort of unrelated healthy adults from the Human Connectome Project ($n = 207$; 83 males, in a range of 22–36 years). The SC represents the group average structural connectome[40].

The Pearson correlation between SC and FC was significantly higher during anesthesia compared with wakefulness ($U = 108.0$, $P = 2.66 \times 10^{-5}$).

We used our framework to simulate wakefulness and propofol anesthesia, modeled as GABA$_A$ receptor potentiation. The firing rates were convolved with a canonical hemodynamic response function to generate simulated BOLD signals (done with the BOLD-monitor integrated in the TVB platform). The calculated FC matrices were correlated with the same reference SC used for the experimental data. Different seeds ($n = 16$) for noise realizations were implemented to address interparticipant variability. The whole-brain model successfully captures the same transition between the simulated data for wakefulness and propofol anesthesia. Despite a larger increase of SC–FC correlation in the case of simulated general anesthesia, the model qualitatively reproduced their relative difference (for a detailed statistical description, refer to Supplementary Information 3.1 and Supplementary Tables 3 and 4).

### State-dependent evoked dynamics—perturbational complexity index.
Conscious and unconscious states exhibit distinct brain responses to external stimuli, with a longer and more widespread propagation of brain activity across the brain during conscious states[41]. This state-dependent response is quantified and clinically study with the perturbational complexity index (PCI), which can distinguish different states of consciousness by quantifying the complexity of electroencephalography responses after a transcranial magnetic stimulation (TMS)-induced direct cortical perturbation[41,42]. Conscious individuals typically demonstrate higher PCI values compared with those during NREM sleep or under general anesthesia (Fig. 6b).

Here, we adopt a similar approach to explore how perturbations propagate through a network of mean fields in the whole-brain model[36] demonstrated that simulated NREM sleep, modeled with increased spike-triggered adaptation in the whole-brain model, was characterized by lower PCI values compared with wake-like states. In this study, we extend this analysis to simulated anesthesia, comparing it with both simulated NREM sleep and wakefulness.

The perturbation was introduced in the whole-brain model as a square wave of amplitude 1 Hz and a duration of 50 ms, applied to the excitatory population of the caudal part of the right middle frontal gyrus. In Fig. 6c, we provide a graphical representation of the spatiotemporal aspects of the signal propagation. The plot illustrates

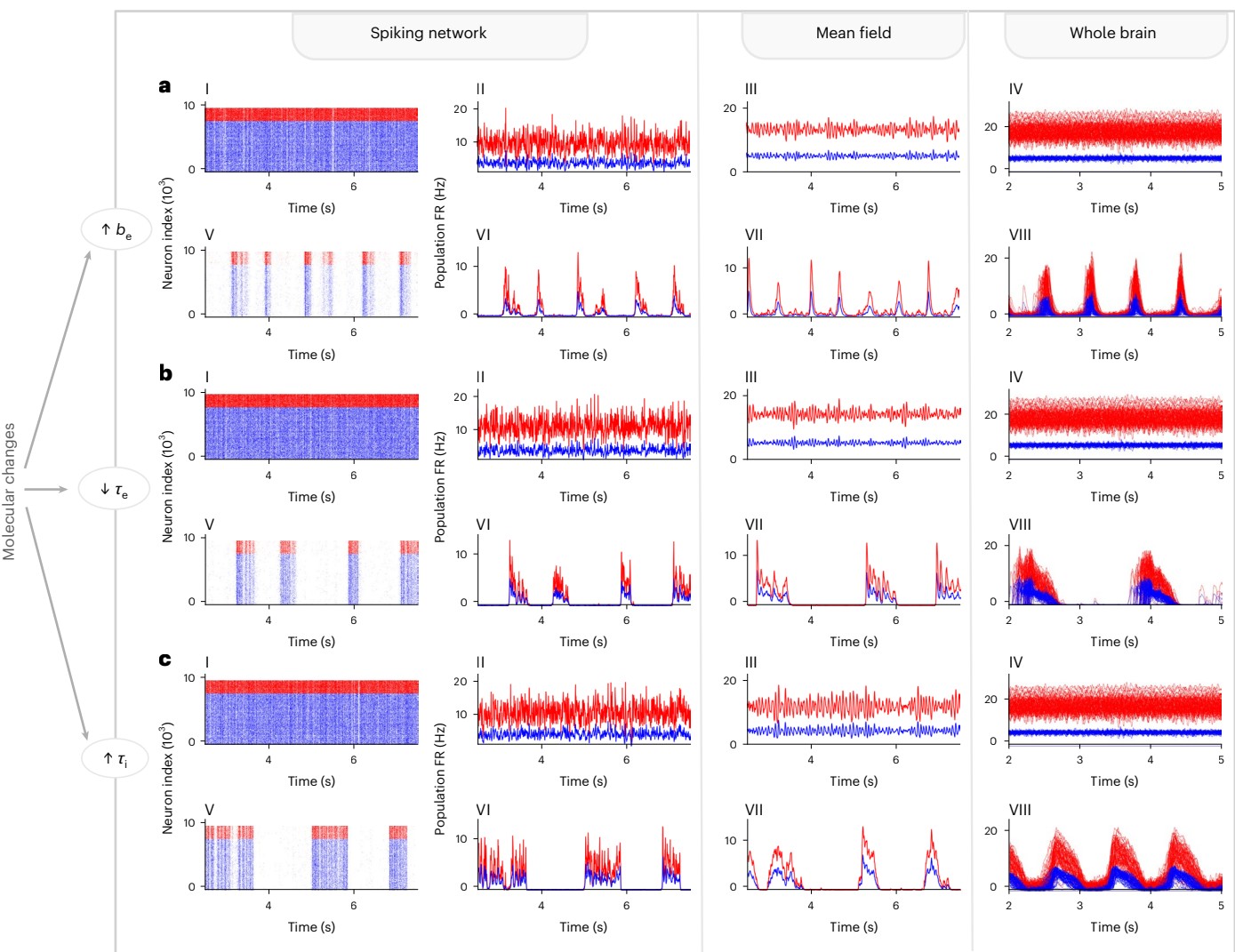

**Fig. 4 | Transition from asynchronous irregular to UP-and-DOWN (UD) dynamics across the different components and scales of our framework: spiking network (single-cell models), mean-field and whole-brain model.** Raster plots (**a**–**c**, I and V) and average firing rate (FR) (**a**–**c**, II and VI) of the excitatory (blue) and inhibitory (red) AdEx neurons in the spiking network, with the corresponding mean-field dynamics (**a**–**c**, III and VII) and whole-brain model (**a**–**c**, IV and VIII). For each of the **a**–**c** cases, a different microscopic parameter ($b_e$, $\tau_e$ and $\tau_i$, respectively) is changed to produce synchronous dynamics. **a**, Increased spike-frequency adaptation current $b_e$ simulating NREM sleep, $b_e = 5$ pA in AI and 120 pA in UD, $\tau_e = \tau_i = 5$ ms in both conditions, $v_{drive} = 0.4$ Hz. In the whole-brain

model, $v_{drive} = 0.315$ Hz for both AI and slow-wave state. **b**, Decreased excitatory synaptic decay, $\tau_e$, mimicking the action of NMDA blockers ($\tau_e = 5$ ms in AI and 3.75 ms in UD, $b_e = 20$ pA in both conditions, $v_{drive} = 0.55$ Hz). In the whole-brain model, $b_e = 5$ pA and $\tau_e = 5$ ms in AI, $b_e = 30$ pA and $\tau_e = 3.75$ ms in synchronous state and $v_{drive} = 0.315$ Hz in both states. **c**, Increased inhibitory synaptic decay, $\tau_i$, mimicking the action of GABA-ergic agents ($\tau_i = 5$ ms in AI and 7.0 ms in UD, $b_e = 10$ pA in both conditions, $v_{drive} = 0.345$ Hz). In the whole-brain model, $b_e = 5$ pA and $\tau_e = 5$ ms in AI, $b_e = 30$ pA and $\tau_i = 7$ ms in synchronous state and $v_{drive} = 0.315$ Hz in both states.

the time each region exceeded the significance activation threshold. The results indicate that, in the case of simulated wakefulness, the signal propagated extensively across the cortex, with regions becoming activated late (depicted in blue). By contrast, during simulated unconscious states, only a few regions surpassed the significance threshold, and these activations occurred primarily shortly after the initial stimulus.

In Fig. 6a, we show the PCI values calculated across 60 trials (using different noise seeds) per condition. We observe that simulated wakefulness is associated with higher PCI values compared with all other unconscious states, namely NREM sleep, NMDA-blocker anesthesia and GABA_A-potentiator anesthesia (see Supplementary Information 3.2 and Supplementary Tables 5–7 for a detailed statistical description).

Figure 6b shows the PCI across various participants, conditions and sessions. The figure compares PCI values measured during wakefulness, NREM sleep and different anesthetic conditions (midazolam,

xenon and propofol). The colors indicate the TMS stimulation intensity (ranging from 80 to 165 V m$^{-1}$), and the different symbols (for instance, triangles, circles and diamonds) represent various TMS targets. The histograms summarize the distribution of PCI values for each condition, showing that wakefulness is characterized by consistently high PCI values across participants. By contrast, during NREM sleep and under anesthesia they are significantly lower, reflecting reduced cortical complexity[42]. Figure 6c shows the time at which each region surpassed the significance threshold after stimulation at $t = 0$, as defined by the PCI calculation algorithm, evaluated and color-coded. The results reveal widespread and temporally sustained signal propagation during wakefulness, but not in the simulated anesthesia and NREM sleep. The stimulated region, located in the caudal part of the right middle frontal gyrus, is marked with a yellow cross. While PCI values in wake-like condition were significantly higher than in the unconscious states ($P < 0.001$), a high degree of overlap was found among the unconscious,

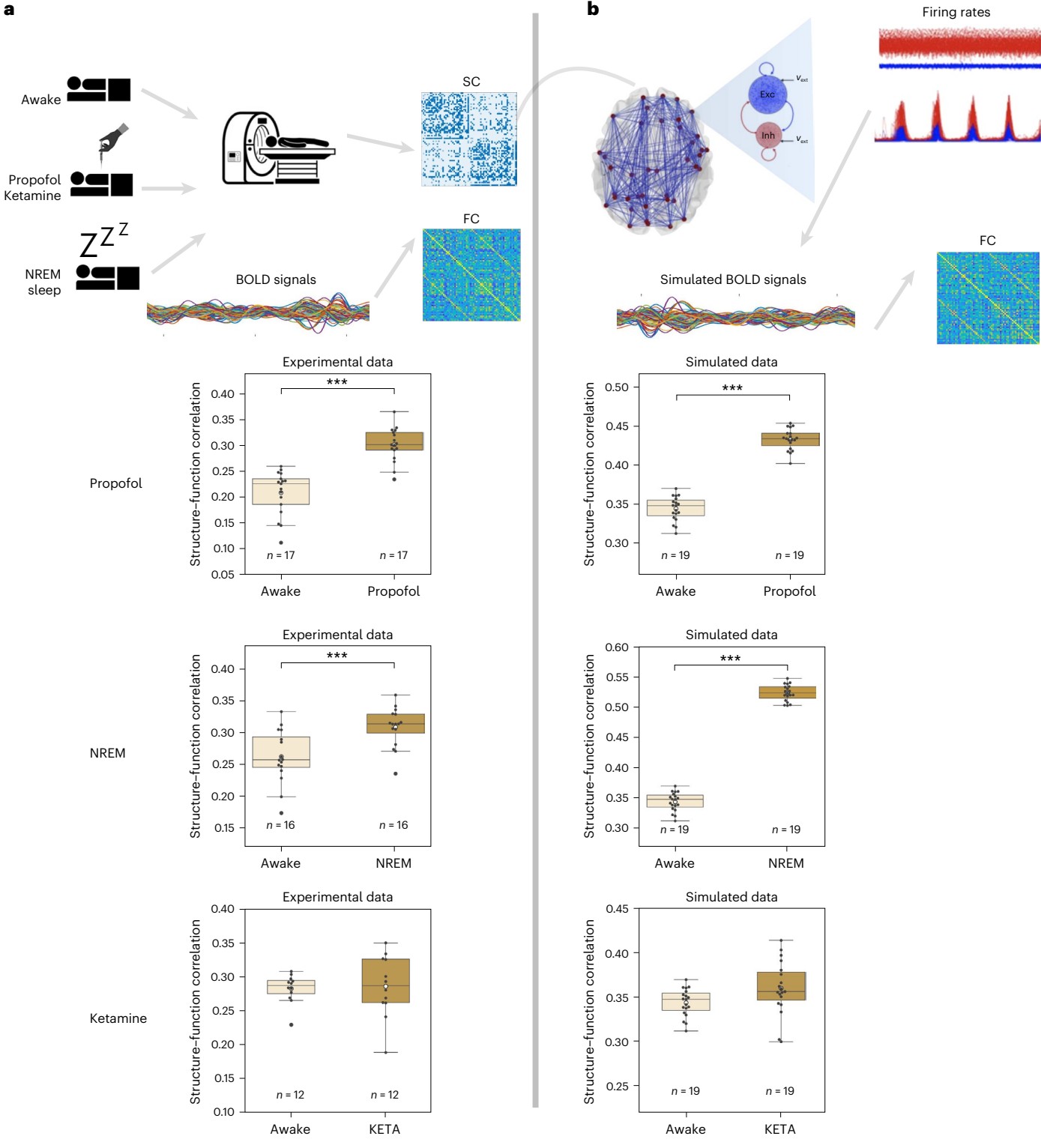

**Fig. 5 | Structure–function correlation is increased under propofol and NREM sleep, but not in the ketamine condition. a**, Resting-state fMRI BOLD signals from different openly available datasets were obtained during wakefulness and under propofol anesthesia, NREM sleep and ketamine anesthesia (KETA) and were used to compute the FC matrix for each participant. We observe a significantly higher correlation between the SC and FC for the case of propofol anesthesia and NREM sleep, but not in the ketamine condition. **b**, The whole-brain model was used to simulate BOLD signals for the wake-like ($b_e = 5$ pA, $\tau_i = 5$ ms, $\tau_e = 5$ ms), propofol anesthesia ($b_e = 30$ pA, $\tau_i = 7$ ms, $\tau_e = 5$ ms), NREM sleep ($b_e = 120$ pA, $\tau_i = 5$ ms, $\tau_e = 5$ ms) and ketamine anesthesia ($b_e = 30$ pA,

$\tau_i = 5$ ms, $\tau_e = 3.75$ ms) conditions (19 different noise realizations). An increased correlation between SC and FC was observed for the case of propofol and NREM sleep, but not in the ketamine condition, similar to the experimental data. ***$P < 0.001$ from Wilcoxon signed-rank test. Middle black line, median; white dot, mean; box limits, first quartile (Q1) and third quartile (Q3) of the data; whiskers, 1.5× interquartile range (IQR); black dots show the SC–FC correlation per participant or noise realization. Exact $P$ values for the experimental and simulated propofol conditions are $1.526 \times 10^{-5}$ and $3.815 \times 10^{-6}$, respectively, and for the experimental and simulated NREM conditions are $2.136 \times 10^{-3}$ and $3.814 \times 10^{-6}$.

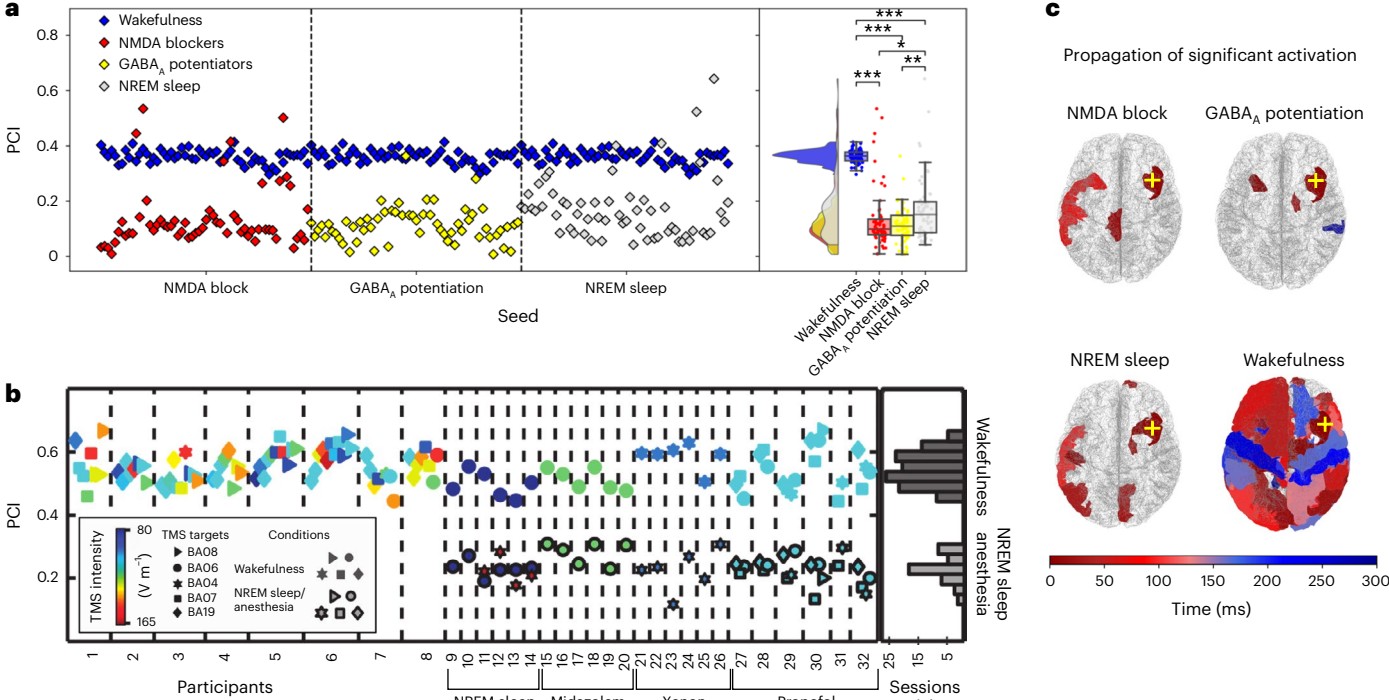

**Fig. 6 | Reduced responsiveness to external stimulus in unconscious states (simulated and empirical). a**, PCI values for the cases of simulated wakefulness ($b_e = 5$ pA, $\tau_e = \tau_i = 5.0$ ms), anesthesia (NMDA blockers ($b_e = 30$ pA, $\tau_e = 3.75$ ms, $\tau_i = 5.0$ ms), GABA$_A$ receptor potentiation ($b_e = 30$ pA, $\tau_i = 7.0$ ms, $\tau_e = 5$ ms) and NREM sleep ($b_e = 120$ pA, $\tau_e = \tau_i = 5.0$ ms). Each point represents a separate noise realization of the model ($n = 60$ simulations per condition, technical replicates). The model was initialized with different noise seeds for each trial, ensuring variability in simulated responses. The horizontal axis corresponds to the noise seed used for the specific simulation, so that each pair of wakefulness–unconscious state is plotted on the same vertical axis; in the right-most panel, the raw jittered data are collapsed to the same vertical axis for each condition; the limits of boxplots include the first quartile (Q1) and third quartile (Q3) of the data, and the middle horizontal lines correspond to the median values; whiskers represent data points within 1.5 times the interquartile range from the quartiles; the half violin plots describe the probability density function; post-hoc Conover tests (two-sided) were performed to assess the statistical significance of the

difference in PCI values between the conditions, and the Holm–Bonferroni method was used to control for multiple comparisons. Exact $P$ values for pairwise comparisons are: GABA$_A$ versus wakefulness ($P = 1.06 \times 10^{-28}$), NMDA block versus wakefulness ($P = 5.73 \times 10^{-27}$), NREM sleep versus wakefulness ($P = 1.2 \times 10^{-18}$) and GABA$_A$ versus NREM sleep ($P = 4.81 \times 10^{-3}$). Mann–Whitney $U$ test statistics ($T$) and effect sizes ($r$) are reported in Supplementary Tables 6 and 7, respectively. (*$P < 0.05$, **$P < 0.01$, ***$P < 0.001$). **b**, Empirical PCI data from human participants showing a clear distinction between wakefulness and NREM sleep and anesthesia conditions. **c**, Spatiotemporal dynamics of applied perturbation propagation. After stimulation, applied at $t = 0$ ms, all regions were evaluated to determine if they surpassed the significance threshold, as defined by the PCI calculation algorithm. The time each region exceeded this threshold is color-coded, showcasing a widespread and temporally sustained signal propagation during wakefulness, but not in the NREM sleep and anesthesia conditions. The yellow cross denotes the stimulated region, corresponding to the caudal part of the right middle frontal gyrus. Panel **b** adapted with permission from ref. 42, AAAS.

with no significant differences detected between the different simulated mechanisms of general anesthesia, although both were lower than NREM sleep.

## Discussion

To demonstrate the application of our framework, we study the effects of common anesthetics at the whole-brain level. Our approach successfully captures the different mechanisms leading to changes in brain states (between conscious and unconscious states) in NREM sleep and anesthesia, highlighting the generality of our approach. Furthermore, we illustrate how our model can explain the transition between dynamical states and the complex interplay between different molecular parameters and mechanisms underlying these transitions. These characteristics make our model suitable for modeling anesthetics in biologically realistic scenarios. The analysis of simulated BOLD signals reveals an increase in structure–function correlation under propofol anesthesia and NREM sleep, but not in the ketamine condition, indicating that FC patterns become more tightly constrained by the underlying structural architecture. This finding aligns with existing literature on consciousness, which consistently shows that structure–function correlation is higher in propofol anesthesia and NREM sleep[43]. Ketamine is a dissociative anesthetic that, at subanesthetic doses, induces an altered state of

consciousness characterized by dissociative symptoms, perceptual distortions and vivid visual and auditory hallucinations. Consistent with existing literature[44], ketamine does not enhance the structure–function correlation, a phenomenon that our model successfully replicates. Although the specific reason behind the different structure–function correlation observed under the simulated ketamine condition may be a combination of different mechanisms and/or neuronal parameters (a detailed analysis is out of the scope of this Article), these results exhibit the capacity of our framework to capture different regimes and subtle emergent differences among patterns of activity.

The reduced responsiveness to external stimuli observed in unconscious states shown in Fig. 6, both in simulated (Fig. 6a,c) and empirical data (Fig. 6b), underscores the robust distinction in PCI levels of consciousness[41,42]. The simulated conditions demonstrate clear reductions in PCI values under simulated anesthesia and NREM sleep conditions, consistent with empirical findings, which show diminished signal complexity in unconscious states. These results highlight that our model predicts the constrained spatiotemporal propagation of perturbations in unconscious conditions.

Previous studies have used the mean-field approach to model the macroscopic effects of general anesthesia on the brain at the macroscale[45], and the action of psychedelic compounds on the

brain[46,47], but without explicitly considering the biophysical details of the molecular action.

Our model holds great potential for clinical applications, where it could be used to predict particular activity patterns as a function of anesthesia depth, such as the level of diminished responsiveness, which may be interesting to explore for intermediate anesthesia levels. Furthermore, our model supports integration with individualized SC data and personalized parameter settings, enhancing its adaptability to patient-specific contexts.

To yield more precise models providing a quantitative comparison with experimental data, it is crucial to accurately reconstruct brain signals (such as electroencephalography, magnetoencephalography or MRI) from mean-field models, as proposed recently[16,48], which will allow a more direct comparison with experiments. This opens also the possibility to estimate model parameters from these brain signals, and their variations due to the action of drugs of altered brain states. Furthermore, although only two neuronal types were considered, our mean-field formalism is compatible with an arbitrary number of neuronal types (see, for example, ref. 11 and ref. 14 for models with three and four neuronal types) and the heterogeneity among neuronal properties can also be incorporated[49]. In addition, it is of great importance to include regional variations, by designing specific models for each brain area, as done recently for noncortical structures[13,14,17]. We hope that this level of biophysical accuracy will bring the model closer to a quantitative account of the fine details of the experimental data and be more precise and specific to investigate their underlying mechanisms.

## Methods

We describe here the details of the multiscale whole-brain model: we start at the cellular level with the introduction of the single-cell model, then we move to the mesoscale with the construction of a neuronal network and the formulation of a mean-field model describing the mesoscopic network activity, and we end at the whole-brain scale using a large network of mean fields interconnected using the anatomical connectome, integrated in the TVB simulator (Fig. 1).

### Single-cell model

The dynamics of the single neuron are based on the AdEx model and are described by the following equations[7]:

$$c_m \frac{dv}{dt} = g_L(E_L - v) + g_L \Delta e^{\frac{v - v_{thr}}{\Delta}} - w + I_{syn} \tag{1}$$

$$\frac{dw}{dt} = -\frac{1}{\tau_w}(a(v - E_L) - w) + b \sum_{t_{sp}} \delta(t - t_{sp}), \tag{2}$$

where $v$ is the membrane potential, $c_m = 200$ pF is the membrane capacitance and $g_L = 10$ nS is the leak conductance. The leakage reversal potential $E_L$ equals $-65$ mV and $-64$ mV for inhibitory and excitatory cells, respectively. The exponential term $\Delta$ has a different strength for the two types of cell (2 mv for excitatory and 0.5 mV for inhibitory cells). $W$ and $I_{syn}$ describe the adaptation and synaptic current respectively. Parameter $a$ describes the subthreshold adaptation, and $b$ describes the spike-triggered adaptation. A spike is generated when the membrane potential exceeds a voltage threshold $v_{thr} = -50$ mV at time $t_{sp}$. The neuron's membrane potential is subsequently reset to a resting voltage $v_{rest} = -65$ mV and fixed to that value for a refractory period $T_{refr} = 5$ ms. The Dirac $\delta$-function indicates that, whenever a neuron fires at time $t_{sp}(k)$, the adaptation current $W$ is incremented by an amount $b$. Based on physiological characteristics[50], the inhibitory cells are modeled as FS neurons, exhibiting no adaptation ($a_i = b_i = 0$). Excitatory neurons are modeled as regular spiking (RS) cells, characterized by lower excitability due to the presence of spiking frequency adaptation ($b_e$ varies in our simulations, $a_e = 0$ nS, and the adaptation time constant $\tau_w = 500$ ms).

### Synaptic model

Each neuron $k$ receives a synaptic current $I_{syn}$, which corresponds to the spiking activity of all presynaptic neurons $j \in \text{pre}(k)$ of neuron $k$. $I_{syn}$ can be decomposed to the input received from excitatory (E) and inhibitory (I) presynaptic spikes, so as

$$I_{syn} = G_{syn}^e(E_e - v) + G_{syn}^i(E_i - v), \tag{3}$$

where $E_e = 0$ mV ($E_i = -80$ mV) is the excitatory (inhibitory) reversal potential and $G_{syn}^e$ ($G_{syn}^i$) the excitatory (inhibitory) synaptic conductance. Synaptic conductances were modeled by a decaying exponential function that sharply increases by a fixed amount $Q_x$, at each spiking time ($t_{sp,j_x}$) of a presynaptic neuron $j_x$:

$$G_{syn}^x(t) = Q_x \sum_{t_{sp,j_x}} \Theta(t - t_{sp,j_x}) e^{-\frac{t - t_{sp,j_x}}{\tau_x}}, \tag{4}$$

where $x$ corresponds to the population type ($x \in \{e, i\}$), $\Theta$ is the Heaviside function, $\tau_x$ is the characteristic decay time of synaptic conductances (varied in our simulations) and $Q_x$ is the quantal conductance ($Q_e = 1.5$ nS, $Q_i = 5$ nS). The sum runs over all the spiking times of excitatory ($t_{sp,j_e}$) or inhibitory ($t_{sp,j_i}$) presynaptic neurons $j_e$ or $j_i$ of neuron $k$.

### Spiking network model

In this work, we considered a network representing a prototypical cortical circuit, specifically a single cortical column[15]. We studied a network of $N = 10^4$ exponential-integrate-and-fire neurons that displayed spike-frequency adaptation[19,50]. The neurons were connected over a topologically random network with a probability of connection between two neurons equal to $P = 5\%$. The network was composed of two populations of inhibitory and excitatory neurons, with the inhibitory neurons consisting the 20% of the whole network size. The input to the network was simulated as a Poissonian input representing the spiking activity of an external population. The spiking-network simulations were performed with the Brian2 Python library[51].

### Mean-field model

The mean-field used here is based on a bottom-up formalism derived from a Master Equation[18], which has recently been extended to include the effects of adaptation[12,19]. The mean-field equations of the system can be written as

$$T \frac{\partial v_\mu}{\partial t} = (F_\mu - v_\mu) + \frac{1}{2} c_{\lambda\eta} \frac{\partial^2 F_\mu}{\partial v_\lambda \partial v_\eta} \tag{5}$$

$$T \frac{\partial c_{\lambda\eta}}{\partial t} = \delta_{\lambda\eta} \frac{F_\lambda(1/T - F_\eta)}{N_\lambda} + (F_\lambda - v_\lambda)(F_\eta - v_\eta)$$
$$+ \frac{\partial F_\lambda}{\partial v_\mu} c_{\eta\mu} + \frac{\partial F_\eta}{\partial v_\mu} c_{\lambda\mu} - 2c_{\lambda\eta} \tag{6}$$

$$\frac{\partial W}{\partial t} = \frac{1}{\tau_w}(a(\mu_V(v_e, v_i, W) - E_L) - W) + b v_e, \tag{7}$$

where $\mu = \{e, i\}$ is the neural population index (excitatory or inhibitory), $v_\mu$ is the mean firing rate of the corresponding population, $c_{\lambda\eta}$ is the covariance between populations ($\lambda$ and $\eta$), $W$ is the mean adaptation and $T$ is the mean-field characteristic time constant.

The function $F_\mu = F_\mu(v_e, v_i, W)$ is the TF of a neuron of type $\mu$, meaning its output firing rate when receiving inhibitory and excitatory inputs with rates $v_e$ and $v_i$, and adaptation level $W$. The TF can be derived following a semi-analytic approach[19], where the output firing rate of a neuron can be written as a function of its mean subthreshold membrane voltage $\mu_V$, its standard deviation $\sigma_V$ and its time correlation decay time $\tau_V$:

$$F = \nu_{\text{out}} = \frac{1}{2\tau_V} \text{Erfc}\left(\frac{V_{\text{thr}}^{\text{eff}} - \mu_V}{\sqrt{2}\sigma_V}\right). \tag{8}$$

where $(\mu_V, \sigma_V, \tau_V)$ are calculated as a function of the input firing rates $(\nu_E, \nu_I)$ and the adaptation intensity $W$ following the equations described in ref. 12. Specifically, the mean membrane potential is calculated as the stationary solution under static conductances driven by the average synaptic input generated by firing rates $(\nu_E, \nu_I)$. This input determines the mean $(\mu_{G_e}, \mu_{G_i})$ and standard deviation $(\sigma_{G_e}, \sigma_{G_i})$ of conductances for excitatory and inhibitory processes. Assuming Poissonian spike statistics (arising from asynchronous irregular dynamics), these values are expressed as

$$\mu_{G_s}(\nu_e, \nu_i) = \nu_s K_s \tau_s Q_s, \tag{9}$$

$$\sigma_{G_s}(\nu_e, \nu_i) = \sqrt{\nu_s K_s \tau_s^2 Q_s}, \tag{10}$$

where $K_s = pN_s$ and $s = \{e, i\}$

The mean input conductance $(\mu_G)$ and the effective membrane time constant $(\tau_m^{\text{eff}})$ are given by

$$\mu_G(\nu_e, \nu_i) = \mu_{G_e} + \mu_{G_i} + g_L, \tag{11}$$

$$\tau_m^{\text{eff}}(\nu_e, \nu_i) = \frac{C_m}{\mu_G}. \tag{12}$$

The average membrane potential for a given adaptation current $w$ is

$$\mu_V(\nu_e, \nu_i, w) = \frac{\mu_{G_e}E_e + \mu_{G_i}E_i + g_L E_L - w}{\mu_G}. \tag{13}$$

The standard deviation $(\sigma_V)$ and time constant $(\tau_V)$ of voltage fluctuations are

$$\sigma_V(\nu_e, \nu_i) = \sqrt{\sum_s K_s \nu_s \frac{(U_s \tau_s)^2}{2(\tau_m^{\text{eff}} + \tau_s)}}, \tag{14}$$

$$\tau_V(\nu_e, \nu_i) = \frac{\sum_s \left(K_s \nu_s (U_s \tau_s)^2\right)}{\sum_s \left(K_s \nu_s \frac{(U_s \tau_s)^2}{\tau_m^{\text{eff}} + \tau_s}\right)}, \tag{15}$$

where $s = \{e, i\}$ and $U_s = Q_s \mu_G (E_s - \mu_V)$.

The $V_{\text{thr}}^{\text{eff}}$ in equation (8) is the phenomenological spike threshold voltage taken as a second-order polynomial

$$V_{\text{eff}}^{\text{thr}}(\mu_V, \sigma_V, \tau_V^N) = P_0 + \sum_{x \in \{\mu_V, \sigma_V, \tau_V^N\}} P_x \left(\frac{x - x_0}{\delta x_0}\right) \\ + \sum_{x,y \in \{\mu_V, \sigma_V, \tau_V^N\}^2} P_{xy} \left(\frac{x - x_0}{\delta x_0}\right)\left(\frac{y - y_0}{\delta y_0}\right), \tag{16}$$

where $\tau_V^N = \frac{\tau_V G_l}{C_m}$ has a non-dimensional quality. The polynomial coefficients of $P$ are determined through a fitting of the TF template to the output firing rate of individual neuron simulations, varying both inhibitory and excitatory inputs. The values for the normalization of the fluctuation regime were set following previous work[12,19]: $\mu_V^0 = -60$ mV, $\sigma_V^0 = 0.004$ mV, $(\tau_V^N)^0 = 0.5$, $\delta\mu_V^0 = 0.001$ mV, $\delta\sigma_V^0 = 0.006$ mV and $\delta(\tau_V^N)^0 = 1$. The fit is performed for each of the cell types included in the network, so in our case for FS and RS cells. As shown in ref. 12, the mean-field predictions work even far from the fitting point of the TF, so in all the examples showcased in this work, we used previously calculated $P$, fitted for similar parameterizations of the two types of cells (Supplementary Table 1).

Taking the first-order equations by disregarding the covariance term dynamics in equation (5), the mean-field model for the excitatory and inhibitory populations representing a cortical volume can be written as

$$T\frac{d\nu_e}{dt} = \mathcal{F}_e(\nu_e + \nu_{\text{aff}}, \nu_i, W) - \nu_e, \tag{17}$$

$$T\frac{d\nu_i}{dt} = \mathcal{F}_i(\nu_e + \nu_{\text{aff}}, \nu_i, W) - \nu_i, \tag{18}$$

$$\frac{dW}{dt} = -\frac{W}{\tau_w} + b\nu_e + \frac{a}{\tau_w}(\mu_V(\nu_e, \nu_i, W) - E_L), \tag{19}$$

where $\nu_{\text{aff}}$ denotes an afferent input to both type of populations which we write as

$$\nu_{\text{aff}}(t) = \nu_{\text{drive}} + \sigma\xi(t), \tag{20}$$

where $\sigma = 3.5$ and $\xi(t)$ denotes an Ornstein–Uhlenbeck (OU) process, of the form

$$d\xi(t) = -\xi(t)\frac{dt}{\tau_{\text{OU}}} + dW_t, \tag{21}$$

where $\tau_{\text{OU}} = 5$ ms is the timescale of the OU process and $dW_t$ is a Wiener process of amplitude one and zero average. The $\nu_{\text{drive}}$ received different values in our simulations depending on the microscopic parameter that was selectively changed.

## Networks of mean-field models

To model whole-brain dynamics, a network of mean fields is defined, where each mean field describes the activity of a brain region. The interactions between the mean fields are excitatory, while the inhibitory connections are preserved solely on the regional level. Extending on the single mean-field equations (equation (17)), now the population activity of each region is given by the following equations:

$$T\frac{d\nu_e(k)}{dt} = \mathcal{F}_e\left[\nu_e^{\text{input}}(k), \nu_i(k), W(k)\right] - \nu_e(k), \tag{22}$$

$$T\frac{d\nu_i(k)}{dt} = \mathcal{F}_i\left[\nu_e^{\text{input}}(k), \nu_i(k), W(k)\right] - \nu_i(k), \tag{23}$$

$$\tau_w \frac{dW(k)}{dt} = -W(k) + b\tau_w \nu_e(k) \\ + a\left(\mu_V(\nu_e(k), \nu_i(k), W(k)) - E_L\right), \tag{24}$$

where $\nu_e(k)$ $(\nu_i(k))$ describes the firing rates of the excitatory (inhibitory) population of the region $(k)$, $W(k)$ is the adaptation of the population, and $\nu_e^{\text{input}}(k)$ is the total excitatory synaptic input that the region receives from the rest of the nodes of the network, given by

$$\nu_e^{\text{input}}(k) = \nu_e(k) + \nu_{\text{aff}}(k) \\ + G\sum_j C_{jk} \nu_e(j, t - \|j - k\|/\nu_c), \tag{25}$$

where the sum runs over all nodes $j$ connected to node $k$, $C_{jk}$ is the connection strength between $j$ and $k$ (and is equal to 1 for $j = k$), and the global coupling factor $G$ rescales the connection strength while maintaining the ratio between different values. The term $\|j - k\|$ is the distance between the nodes $j$ and $k$, while $\nu_c$ is the speed with which the signal propagates along the axis so that the model accounts for the delay of axonal propagation. Here, $\nu_{\text{aff}}$ is defined as in equation (20).

A cortical parcellation of 68 regions was used. The connection strengths and tract lengths between the nodes were provided by the Berlin empirical data processing pipeline, based on human tractography methods[52]. We performed the simulations using the TVB platform, leveraging its capabilities to model and simulate large-scale brain networks. TVB offers a useful tool to study the dynamics of the entire brain through the construction of a network defined by its large-scale SC and the mesoscopic models that describe each node's intrinsic dynamics. This approach allows us to simulate and analyze the complex interactions and emergent properties of the brain's functional dynamics at both the mesoscopic and macroscopic levels.

### Data for structural–functional connectivity analysis

**Simulated BOLD signals.** The simulated BOLD signal was generated through the dedicated BOLD monitor provided by TVB software. The simulated mean field time series of the excitatory population (downsampled at 250 Hz) of each node were convolved with the hemodynamic response function (described by the first-order Volterra kernel of the Ballon Windekessel model[53]). This signal was finally downsampled to the frequency matching the repetition time (TR) of the fMRI series. For the simulations of the human brain, the human connectome described in 'Networks of mean-field models' section was used.

**Human fMRI data.** Details can be found in ref. 54.

### Evoked activity and PCI calculation

The PCI, as proposed by Casali et al.[42], is the normalized Lempel–Ziv complexity of the evoked spatiotemporal patterns of cortical activation $SS(x, t)$ and was computed to capture the complexity of the signal propagation after a stimulation. Evoked potentials were generated using the whole-brain model, where a square wave stimulus was applied on the firing rates of the excitatory population on a single node (corresponding to the right premotor cortex). Sixty different trials, each with a different stimulus onset and noise realization, were performed where the analysis concerns an interval of 300 ms before and after the stimulus. The firing rates of the excitatory populations were first normalized using the mean and standard deviation of the prestimulus interval. To assess statistical significance, a bootstrap procedure was used to generate a null distribution of surrogate values. This was done by repeatedly shuffling the prestimulus signals. A significance threshold was then determined from the null distribution. Finally, the normalized poststimulus firing rates were compared with this threshold, resulting in a binary map of significant vectors $S(t)$ for each trial. The Lempel–Ziv complexity $LZ(S)$ was calculated for each of these vectors[36]. Finally the PCI is expressed as the ratio of $\frac{LZ(S)}{H(S)}$ where $H(S)$ is the spatial source entropy defined in equation (26), being $p_0(S)$ the fraction of elements equal to 0 contained in $S$ and $p_1(S)$ the fraction of elements equal to 1.

$$H(S) = -p_0(S)\log_2(p_0(S)) \\ -p_1(S)\log_2(p_1(S)). \tag{26}$$

### Reporting summary

Further information on research design is available in the Nature Portfolio Reporting Summary linked to this article.

### Data availability

The experimental data used for the structure–function correlation analysis for propofol can be found in refs. 55,56, for ketamine in refs. 57,58 and for sleep in ref. 59. The structural connectivity matrix can be found in ref. 40. The connectivity data used for the whole-brain simulations is available via GitHub at https://github.com/mariasacha/paper_pipeline_hub/tree/master/TVB/tvb_model_reference/data/QL_20120814. Source data are provided with this paper.

### Code availability

The code source of all simulations shown in this Article is available via GitHub at https://github.com/mariasacha/paper_pipeline_hub/ and via Zenodo at https://doi.org/10.5281/zenodo.15035287 (ref. 60). Our code makes use of The Virtual Brain library, available at https://www.thevirtualbrain.org/tvb/zwei/home (version 2.9).

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

## Acknowledgements

Research supported by the CNRS, the Agence Nationale de La Recherche (ANR grants ImpactCom and BrainAct) and the European Union (Human Brain Project H2020-785907, H2020-945539, Virtual Brain Twin project 101137289). We thank E. Leveque for her help in the simulated BOLD-signal analysis. Code from https://gitlab.ebrains.eu/kancourt/tvb-adex-showcase3-git was used as a foundation for the TVB simulations. Therefore, we thank L. Kusch, B. Hazal Yalçınkaya, T.-A. Nghiem, D. Aquilhue-Llorens and J. Goldman for their contributions to the open-source community. We also thank the ENIGMA consortium for providing the structural connectome data used in generating Fig. 5, Z. Huang for making the propofol and ketamine data publicly available, and Y. Sanz-Perl and E. Tagliazucchi for sharing the data from sleep participants.

## Author contributions

M.S., F.T. and A.D. conceived the model, M.S., F.T. and R.C. performed the analysis and simulations. All authors contributed to the interpretation of the results and co-wrote the final the paper. A.D. supervised the project.

## Competing interests

The authors declare no competing interests.

## Additional information

**Correspondence and requests for materials** should be addressed to Alain Destexhe.

# Reporting Summary

## Statistics

For all statistical analyses, confirm that the following items are present in the figure legend, table legend, main text, or Methods section.

| n/a | Confirmed | |
|---|---|---|
| ☐ | ☒ | The exact sample size (*n*) for each experimental group/condition, given as a discrete number and unit of measurement |
| ☒ | ☐ | A statement on whether measurements were taken from distinct samples or whether the same sample was measured repeatedly |
| ☐ | ☒ | The statistical test(s) used AND whether they are one- or two-sided<br>*Only common tests should be described solely by name; describe more complex techniques in the Methods section.* |
| ☒ | ☐ | A description of all covariates tested |
| ☐ | ☒ | A description of any assumptions or corrections, such as tests of normality and adjustment for multiple comparisons |
| ☐ | ☒ | A full description of the statistical parameters including central tendency (e.g. means) or other basic estimates (e.g. regression coefficient) AND variation (e.g. standard deviation) or associated estimates of uncertainty (e.g. confidence intervals) |
| ☐ | ☒ | For null hypothesis testing, the test statistic (e.g. *F*, *t*, *r*) with confidence intervals, effect sizes, degrees of freedom and *P* value noted<br>*Give P values as exact values whenever suitable.* |
| ☒ | ☐ | For Bayesian analysis, information on the choice of priors and Markov chain Monte Carlo settings |
| ☒ | ☐ | For hierarchical and complex designs, identification of the appropriate level for tests and full reporting of outcomes |
| ☐ | ☒ | Estimates of effect sizes (e.g. Cohen's *d*, Pearson's *r*), indicating how they were calculated |

*Our web collection on statistics for biologists contains articles on many of the points above.*

## Software and code

Policy information about availability of computer code

| | |
|---|---|
| Data collection | The code used for the simulations of this study was developed by our group. The code source of all simulations shown in this article is available online in https://github.com/mariasacha/paper_pipeline_hub/ where the exact versions of the packages used can be found in the requirements.txt. Our code use the Virtual Brain https://www.thevirtualbrain.org/tvb/zwei/home (version 2.9) |
| Data analysis | The code used for the simulations of this study was developed by our group. The code source of all simulations shown in this article is available online in https://github.com/mariasacha/paper_pipeline_hub/ where the exact versions of the packages used can be found in the requirements.txt. Our code use the Virtual Brain https://www.thevirtualbrain.org/tvb/zwei/home (version 2.9) |

For manuscripts utilizing custom algorithms or software that are central to the research but not yet described in published literature, software must be made available to editors and reviewers. We strongly encourage code deposition in a community repository (e.g. GitHub). See the Nature Portfolio guidelines for submitting code & software for further information.

## Data

Policy information about availability of data

All manuscripts must include a data availability statement. This statement should provide the following information, where applicable:
- Accession codes, unique identifiers, or web links for publicly available datasets
- A description of any restrictions on data availability
- For clinical datasets or third party data, please ensure that the statement adheres to our policy

All the experimental data used in this article in openly available and have been previously published. For the structure-function correlation analysis for Propofol can be found in (Jang et al., 2024), for Ketamine (Huang et al., 2023) and Sleep in (Stikvoort et al., 2024). The structural connectivity matrix can be found in (Larivière et al.,2021). The connectivity data used for the whole brain simulations can be found in https://github.com/mariasacha/paper_pipeline_hub/tree/master/TVB/tvb_model_reference/data/QL_20120814. Source data for Figures 3, 4, 5, and 6 are available with this manuscript.

## Human research participants

Policy information about studies involving human research participants and Sex and Gender in Research.

| | |
|---|---|
| Reporting on sex and gender | No data from human were collected during this study |
| Population characteristics | No data from human were collected during this study |
| Recruitment | No data from human were collected during this study |
| Ethics oversight | No data from human were collected during this study, and no ethical overisght was needed |

Note that full information on the approval of the study protocol must also be provided in the manuscript.

# Field-specific reporting

Please select the one below that is the best fit for your research. If you are not sure, read the appropriate sections before making your selection.

☒ Life sciences      ☐ Behavioural & social sciences      ☐ Ecological, evolutionary & environmental sciences

For a reference copy of the document with all sections, see nature.com/documents/nr-reporting-summary-flat.pdf

# Life sciences study design

All studies must disclose on these points even when the disclosure is negative.

| | |
|---|---|
| Sample size | For the simulations that were compared with experimental data, different seeds were used to introduce variability in the results. For the case of structural and functional connectivity analysis, the number of seeds was selected in order to match the number of available human samples. For the case of PCI analysis, we chose a sufficiently large number of seeds that was permitted by the restrictions of time and computational resources |
| Data exclusions | No data were excluded |
| Replication | To ensure replication, all the scripts that were used for the simulations are accessible to the readers |
| Randomization | Not relevant to our study that involves simulations |
| Blinding | Not relevant to our study that involves simulations |

# Reporting for specific materials, systems and methods

We require information from authors about some types of materials, experimental systems and methods used in many studies. Here, indicate whether each material, system or method listed is relevant to your study. If you are not sure if a list item applies to your research, read the appropriate section before selecting a response.

## Materials & experimental systems

| n/a | Involved in the study |
|---|---|
| ☒ | ☐ Antibodies |
| ☒ | ☐ Eukaryotic cell lines |
| ☒ | ☐ Palaeontology and archaeology |
| ☒ | ☐ Animals and other organisms |
| ☒ | ☐ Clinical data |
| ☒ | ☐ Dual use research of concern |

## Methods

| n/a | Involved in the study |
|---|---|
| ☒ | ☐ ChIP-seq |
| ☒ | ☐ Flow cytometry |
| ☒ | ☐ MRI-based neuroimaging |

