## [Peer Review File · Nature Computational Science]

A computational approach to evaluate how molecular mechanisms impact large-scale brain activity

Corresponding Author: Dr Alain Destexhe

Version 0:

Decision Letter:

Dear Dr Destexhe,

Thank you very much for your enquiry about submitting a manuscript to Nature Computational Science.

I've now had a chance to discuss your work with my colleagues, and although we think that it sounds very interesting, we are still uncertain as to the degree to which the study shows applications of the work along with use cases.

Therefore, we would like to invite you to submit the full manuscript to Nature Computational Science so that we can examine the data before deciding whether to send the paper out to review.

If this is acceptable to you, you can submit the complete manuscript using the link below:

Link Redacted

If you have any questions, please feel free to contact me.

Best regards,

Ananya Rastogi, PhD
Senior Editor
Nature Computational Science

Version 1:

Decision Letter:

**** Please ensure you delete the link to your author homepage in this e-mail if you wish to forward it to your co-authors. ****

Dear Dr Destexhe,

Thank you again for submitting your manuscript "A computational approach to evaluate how molecular mechanisms impact large-scale brain activity". I am pleased to tell you that we are sending your paper out for formal peer review. Before we can do so, please read the below carefully as we require a few documents.

If you have not done so already, please alert us to any related manuscripts from your group that are under consideration or in press at other journals, or are being written up for submission to other journals (see <https://www.nature.com/authors/policies/duplicate.html> for details).

We are asking all corresponding authors of primary research articles to complete an Editorial Policy Checklist that verifies compliance with all required editorial policies. The form should be completed and returned within 48 hours, if you have not already done so. Please note that the form is a dynamic 'smart pdf' and must therefore be downloaded and completed in Adobe Reader. We will then flatten them for ease of use by the reviewers. If you would like to reference the guidance text as

you complete the template, please access these flattened versions at <https://www.nature.com/authors/policies/availability.html>

Editorial Policy Checklist: <https://www.nature.com/documents/nr-editorial-policy-checklist.pdf>

We also ask that you complete a reporting summary that collects information on experimental design:
Reporting summary: <https://www.nature.com/documents/nr-reporting-summary.pdf>

In addition, as your paper relies on code that is central to the main claims, it should be shared with the referees during the peer review process and publicly at least on publication. If there are restrictions to public sharing on publication, please discuss them with us.

You have selected to use our support service for code submission, peer review and publication using the platform Code Ocean. As part of this service, you will be assisted to create a compute capsule that will be used for peer review and eventual publication of the code associated with your paper.

Please use the link below to log in or sign up to Code Ocean. Your manuscript will be linked to your Code Ocean account, and will appear in the metadata of your compute capsule. Please navigate to the Metadata tab and select this manuscript in the 'Associated Publication' section.

Log in to Code Ocean to complete your code submission request: <https://codeocean.com/signup/nature?token=7553b670b94a40459c377c3d4ffd7281>

More information about setting up the capsule can be found here: <https://docs.codeocean.com/user-guide/>

If you have any other questions, please email support@codeocean.com. Where possible, please copy us in all communication with Code Ocean.

To improve transparency in authorship we are requesting that all authors identified as 'corresponding author' on published papers create and link their Open Researcher and Contributor Identifier (ORCID) with their account on the Manuscript Tracking System (MTS), prior to acceptance. ORCID helps the scientific community achieve unambiguous attribution of all scholarly contributions. You can create and link your ORCID from the home page of the MTS by clicking on 'Modify my Springer Nature account'. For more information please visit <http://www.springernature.com/orcid>.

Finally, we encourage you to share a preprint of the original submitted version of your paper so as to minimize delays in communicating your research findings; benefits of preprints include early visibility, and citations (<https://www.natureindex.com/news-blog/preprints-boost-article-citations-and-mentions>) and demonstration of research progress. You may want to consider the multidisciplinary Research Square preprint platform (<https://www.researchsquare.com/browse>), provided by our partner Research Square, where your preprint will be publicly available with a citable DOI under a CC-BY license. You are of course free to use a discipline-specific preprint platform of your choice. More information about our preprint policy can be found in the following link: <https://www.nature.com/nature-research/editorial-policies/preprints-and-conference-proceedings#preprints>

Please use the following link to submit the required checklists; please also resubmit your original manuscript files, or revised versions of them as a result of filling out the checklists:

Link Redacted

** This url links to your confidential homepage and associated information about manuscripts you may have submitted or be reviewing for us. If you wish to forward this e-mail to co-authors, please delete this link to your homepage first. **

Please note that Nature Computational Science implements transparent peer review of original research manuscripts, in which we publish the reviewer comments to the authors, author rebuttal letters and editorial decision letters as a supplementary peer review file, if the author agrees at the point of acceptance. This will apply to new manuscripts submitted on or after 17th Feb 2021. Upon author request, confidential information and data can be removed from the reviewer reports and rebuttal letters prior to publication. For more information, please refer to our [FAQ page](https://www.nature.com/documents/nr-transparent-peer-review.pdf).

Best regards,

Ananya Rastogi, PhD
Senior Editor
Nature Computational Science

Version 2:

Decision Letter:

**** Please ensure you delete the link to your author homepage in this e-mail if you wish to forward it to your co-authors. ****

Dear Dr Destexhe,

Your manuscript "A computational approach to evaluate how molecular mechanisms impact large-scale brain activity" has now been seen by 3 referees, whose comments are appended below. You will see that while they find your work of interest, they have raised points that need to be addressed before we can make a decision on publication.

The referees' reports seem to be quite clear. Naturally, we will need you to address ***all*** of the points raised.

While we ask you to address all of the points raised, the following points need to be substantially worked on:

- The propagation of the NMDA and GABA receptor properties from the spiking network to the mean field model to the virtual brain model is not explained. Please correct this.
- Please explain how were the effects at the microscale incorporated in the macroscale model.
- It would be useful for the reader to have a discussion on how one could use this framework to systematically estimate model parameters.
- Please analyse how well the model can capture whole-brain dynamics in terms of Functional connectivity, a traditional benchmark for the prediction power of the model.
- the paper should better define the spiking network.
- In the Discussion, add a proper comparative analysis.
- Please annotate the code..

In addition to these points, it would also be beneficial to address the following concerns:

- Critical information about Methods is missing and there is a limited discussion. Please add this.
- The explanation of how molecular mechanisms propagate across scales should be extended to include all details.

Please use the following link to submit your revised manuscript and a point-by-point response to the referees' comments (which should be in a separate document to any cover letter):

Link Redacted

**** This url links to your confidential homepage and associated information about manuscripts you may have submitted or be reviewing for us. If you wish to forward this e-mail to co-authors, please delete this link to your homepage first. ****

To aid in the review process, we would appreciate it if you could also provide a copy of your manuscript files that indicates your revisions by making use of Track Changes or similar mark-up tools. Please also ensure that all correspondence is marked with your Nature Computational Science reference number in the subject line.

In addition, please make sure to upload a Word Document or LaTeX version of your text, to assist us in the editorial stage.

To improve transparency in authorship, we request that all authors identified as 'corresponding author' on published papers create and link their Open Researcher and Contributor Identifier (ORCID) with their account on the Manuscript Tracking System (MTS), prior to acceptance. ORCID helps the scientific community achieve unambiguous attribution of all scholarly contributions. You can create and link your ORCID from the home page of the MTS by clicking on 'Modify my Springer Nature account'. For more information please visit www.springernature.com/orcid.

We hope to receive your revised paper within three weeks. If you cannot send it within this time, please let us know.

Best regards,

Ananya Rastogi, PhD
Senior Editor
Nature Computational Science

Reviewers comments:

Reviewer #1 (Remarks to the Author):

This paper reports an ordered sequence of models, from micro to meso to macro scale, and uses it to demonstrate a

biological property of the brain, i.e., the state transitions occurring with deep anesthesia. The microscale, mesoscale, and macroscale models have been previously described in great detail in different papers, some also by the same authors. The Authors place an extended methodological description in most of the paragraphs of the result section, implying that this is indeed a paper on methods. Or better on a workflow, as they insist on the generality of their approach in several instances. Although the contribution to the bottom-up modelling strategy is commendable, there are some points to consider.

(1) The propagation of the NMDA and GABA receptor properties from the spiking network to the mean field model to the virtual brain model is not explained. For example, the mean field model does not show any parameters directly related to NMDA or GABA receptors. The parameters of the mean field model that are closest to NMDA and GABA receptor currents, τ_e and τ_i , do not appear in any of the equations reported in the Methods. This explanation is crucial but is missing. An explanation here is needed, also for the sake of reproducibility.

(2) The paper is about the propagation of molecular-level properties to higher scales, but in the intro the impact of pharmaceutical compounds is declared. The pharmacological issue here is a reflection of the simulation of changes in membrane receptor functions. There is no specific workflow to simulate drugs acting on the receptors though. This should be clarified.

(3) The authors simulate state transition of cortical activity from wakefulness to sleep. The change is evident in the spiking network, in the mean field model, and in the virtual brain model. This is a nice *in silico* experiment. However, the discussion of results shown in Fig. 5 and 6 is very limited.

- What about the firing rate of the E and I neurons? By the way, in the mean field model two neurons represent more than 50 types of neurons of a cortical column that contains more than 30000 neurons as a whole. This should be explained. Any comparison between mean field model firing rates and those of the spiking network when anesthesia states are simulated?
- Fig. 6 looks quite rich of information, but it is not described or commented beyond the figure legend. It must be described in order to make it understandable.

Egidio D'Angelo

Reviewer #2 (Remarks to the Author):

In this manuscript, the authors use a mean field based computational framework to evaluate pharmaceutical interventions on whole brain activity. They use a very interesting bottom-up approach from single cell models to a whole brain network model so that the effect of pharmaceuticals can be introduced at the single cell microscale model parameters while observing the effects on a macroscale. This approach is innovative and significant for designing therapeutics specially in human subjects where we can only measure at the macroscopic level while the intervention works at the microscopic level. Using this approach, they show that anesthetics targeting GABA and NMDA receptors modelled as changes in the cellular membrane time constant can switch whole brain activity to generalized slow wave patterns as observed experimentally. This is an overall very well written paper with clear explanations. I have the following doubts regarding the methods that was not very clear to me and would appreciate some clarifications:

1) How are the single cell model parameters related to the mean field model parameter? In other words, how were the effects at the microscale incorporated in the macroscale model. For example, is the mean field time constant T related in some functional way to the τ s that were manipulated to simulate effects of anesthetics. A description of this would be useful as this is the most salient feature of this paper.

2) This is a forward model and produces output data that matches with experimental data. It would be useful for the reader to have a discussion on how one could use this framework to systematically estimate model parameters such as the τ s if one has the experimental data. For example, if we had the EEG/MEG recordings, can we use this data to find what model parameters are necessary to produce this observed brain recording. This would be in my opinion crucial for designing therapeutics and would take this approach one step closer to translation.

Reviewer #2 (Remarks on code availability):

I have browsed through the code and a readme file is present with instructions on which file to use. However, I have not run the code.

Reviewer #3 (Remarks to the Author):

Review for the Nature Computational Paper

In this paper, the authors have devised a framework that can build a whole brain model from a neuronal level, which can

incorporate the dynamics of ion channels at the cellular level, spiking patterns at the microscale level and structural information from tractography data at the mesoscale level. This unique framework has four components – a spiking neuron model at a single neuron level, a network of spiking neuron models, a low dimensional mean field model incorporating the information of spiking neurons, and a whole brain model with the mean-field model constrained by the structural connectivity. The current framework can be a valuable tool for deciphering the impact of the aberration of ion channels on neurological disorders or unconscious states at the macroscale or whole brain level. One of the applications of this model has been shown in the case of anaesthesia, where the pharmaceutical interval by Ketamine and Propofol is investigated both at the neural level (spiking pattern) and macroscale level (structure-function) relationship and with perturbation.

The paper has exciting modelling aspects and has the potential to understand the impact of aberrations in neural dynamics at the whole brain level. The paper has several positive points, including the revised new methodology to make a more biologically plausible whole brain model from a high dimensional spiking network model to a low dimensional mass model (bottom-up approach), the rigorous nature of the analysis, usage of the schematic diagram to discuss the underlying biological events (figure 1 and 2) and illustrating the application of the model to decipher the biological phenomena during the anaesthesia.

The paper is suitable for publication, however the detailed revisions is needed.

However, some parts of the paper are unclear to the reviewer, which needs to be clarified.

Firstly, the paper investigates the structure-function relationship in reference and anaesthesia cases. However, it does not reveal how well the model can capture whole-brain dynamics in terms of Functional connectivity, a traditional benchmark for the prediction power of the model. The authors can refer to the work of Luppi et al. (<https://www.nature.com/articles/s42003-022-03330-y>), where the model was tested with the Functional connectivity dynamics (FCD). A similar analysis can investigate the precision of the model.

Secondly, the paper could better define the spiking network. It would be beneficial to include a description in the methods section regarding how the network was constructed and the rationale behind using 80% pyramidal (excitatory) neurons and 20% interneurons (inhibitory). A reference may suffice in this instance.

Thirdly, the model investigates the underlying dynamics of the model by analyzing the excitatory firing pattern with the transfer function associated with it, and the transition state was shown with the loss of a stable self-sustained state and the emergence of an up-down state. However, it's difficult to follow this in section 2.6. "Emergence of UP and Down states....." a basic description is needed to make the paper self-sufficient. Some of the arguments from the earlier work by Di Volo et al. (2019) can be added to clarify the rationale behind referring to it as an 'Up and Down' state and the bistable behaviour of the model. As it is one of the central arguments of the work reflecting the impact of the parameters describing the interventions of propofol, it should be stated clearly.

Fourthly, the frequent references to the NREM sleep stage make the results section somewhat confusing. The comparison and analysis should either be discussed separately or mentioned in the discussion section. It is challenging to distinguish outcomes pertaining to anaesthesia cases from those related to sleep stages.

Fifth, in the figures given in Fig. 6 A and B, it is impossible to follow the scattered plots for PCI (Perturbational Complexity Index) and make inferences from them. The authors can remove it by replacing it with a simple box plot with statistical analysis, which is enough to showcase the difference.

Sixth, In the Discussion, the paper lacks the proper comparative analysis. A brief discussion about the superiority of the model over the existing ones (i.e. Luppi et al., Deco et al. (2018), referenced in the paper but not discussed) can be described in terms of what can be inferred from the model, which is not possible from the other models. These arguments will make a strong case for this modelling approach.

Seventh, the paper needs a parameter table to track the changes in parameters at a glance for different cases. Some of the parameters are not even available, like- V_c in the equation in line number 684. It looks like it is a velocity parameter that incorporates the delay information. However, its importance has been shown in earlier mean field models like- Castaldo et al. (2023) <https://doi.org/10.1016/j.neuroimage.2023.120236>.

Lastly, the provided code is not annotated fully. It's difficult to follow it seamlessly and produce the results given in the paper. Authors can add a parameter file or clearly mention the parameters involved in the paper to produce the results.

Minor revisions that must be addressed.

1. In the summary of Figure 4, the authors mention UD. What is it?
2. In line 408, the authors talked about correlating SC with FC. The term "Reference SC" is used. Please clarify it. Is it a mean SC of all the individuals or an individual SC? Is the model a personalized model meaning a separate SC used to compute the FC of each person?
3. The chronology of numbering for the equations is not followed; some are numbered, and some are not. Please enumerate them.
4. In line 743, the phrase "significant vectors..." sounds vague. Rephrase it.

Version 3:

Decision Letter:

Our ref: NATCOMPUTSCI-24-1171C

20th February 2025

Dear Dr. Destexhe,

Thank you for submitting your revised manuscript "A computational approach to evaluate how molecular mechanisms impact large-scale brain activity" (NATCOMPUTSCI-24-1171C). It has now been seen by the original referees and their comments are below. The reviewers find that the paper has improved in revision, and therefore we'll be happy in principle to publish it in Nature Computational Science, pending minor revisions to satisfy the referees' final requests and to comply with our editorial and formatting guidelines.

TRANSPARENT PEER REVIEW

Nature Computational Science offers a transparent peer review option for original research manuscripts. We encourage increased transparency in peer review by publishing the reviewer comments, author rebuttal letters and editorial decision letters if the authors agree. Such peer review material is made available as a supplementary peer review file. **Please remember to choose, using the manuscript system, whether or not you want to participate in transparent peer review.**

Thank you again for your interest in Nature Computational Science. Please do not hesitate to contact me if you have any questions.

Sincerely,

Ananya Rastogi, PhD
Senior Editor
Nature Computational Science

ORCID

Reviewer #1 (Remarks to the Author):

All my questions have been answered and I think the paper is much improved. I do not have further issues.

Reviewer #2 (Remarks to the Author):

My concerns have been satisfactorily addressed

Reviewer #3 (Remarks to the Author):

In the first round of reviews, I suggested improvements to specific sections of the paper and requested further explanations of certain methodologies. The authors responded well by providing additional details that clarified their arguments. Below, I outline my evaluation of the revisions.

For the first point, the authors explained the model's accuracy by emphasizing that the model's objective is not just to reconstruct the FC but to find a correlation between SC and FC. They also included the functional connectivity density

(FCD) for various cases based on my suggestions. This provides an opportunity to compare the model's performance in healthy and different anaesthetic conditions. However, the KS distance is large, a meaningful insight can be gained from the computational study.

Secondly, the authors have defined the spiking network separately, complete with proper citations.

Thirdly, the up-down state and bi-stability are well-defined now and easy to pursue.

In response to my fourth comment, the authors have modified the results and discussions to distinguish several cases.

For my fifth comment, the clarification for depicting Fig. 6A. is noted.

For the sixth point, a good discussion is presented at the end of the paper.

The authors also added Table 2, which is essential for clarifying the confusion about the parameter values. Now, the paper looks complete.

The codes are checked and found to be well documented, and apart from minor issues, e.g. file names, there are no significant issues in the simulation.

Lastly, all minor points have been addressed satisfactorily.

I appreciate the efforts made by the authors in revising the manuscript. I am glad that the authors found my suggestions valuable. I recommend the article for publication.

Version 4:

Decision Letter:

Dear Dr Destexhe,

We are pleased to inform you that your Article "A computational approach to evaluate how molecular mechanisms impact large-scale brain activity" has now been accepted for publication in Nature Computational Science.

Once your manuscript is typeset, you will receive an email with a link to choose the appropriate publishing options for your paper and our Author Services team will be in touch regarding any additional information that may be required.

Acceptance of your manuscript is conditional on all authors' agreement with our publication policies (see <https://www.nature.com/natcomputsci/for-authors>). In particular your manuscript must not be published elsewhere and there must be no announcement of the work to any media outlet until the publication date (the day on which it is uploaded onto our web site).

Before your manuscript is typeset, we will edit the text to ensure it is intelligible to our wide readership and conforms to house style. We look particularly carefully at the titles of all papers to ensure that they are relatively brief and understandable.

Once your manuscript is typeset, you will receive a link to your electronic proof via email with a request to make any corrections within 48 hours. If, when you receive your proof, you cannot meet this deadline, please inform us at rjsproduction@springernature.com immediately.

If you have queries at any point during the production process then please contact the production team at rjsproduction@springernature.com.

You may wish to make your media relations office aware of your accepted publication, in case they consider it appropriate to organize some internal or external publicity. Once your paper has been scheduled you will receive an email confirming the publication details. This is normally 3-4 working days in advance of publication. If you need additional notice of the date and

time of publication, please let the production team know when you receive the proof of your article to ensure there is sufficient time to coordinate. Further information on our embargo policies can be found here:
<https://www.nature.com/authors/policies/embargo.html>

We welcome the submission of potential cover material (including a short caption of around 40 words) related to your manuscript; suggestions should be sent to Nature Computational Science as electronic files (the image should be 300 dpi at 210 x 297 mm in either TIFF or JPEG format). We also welcome suggestions for the Hero Image, which appears at the top of our [home page](http://www.nature.com/natcomputsci); these should be 72 dpi at 1400 x 400 pixels in JPEG format. Please note that such pictures should be selected more for their aesthetic appeal than for their scientific content, and that colour images work better than black and white or grayscale images. Please do not try to design a cover with the Nature Computational Science logo etc., and please do not submit composites of images related to your work. I am sure you will understand that we cannot make any promise as to whether any of your suggestions might be selected for the cover of the journal.

Best regards,
Fernando

--
Fernando Chirigati, PhD
Chief Editor, Nature Computational Science
Nature Portfolio

P.S. Click on the following link if you would like to recommend Nature Computational Science to your librarian: <https://www.springernature.com/gp/librarians/recommend-to-your-library>

** Visit the Springer Nature Editorial and Publishing website at <http://editorial-jobs.springernature.com> for more information about our career opportunities. If you have any questions please click [here](mailto:editorial.publishing.jobs@springernature.com). **

Nature Computational Science manuscript
NATCOMPUTSCI 24 1171B

December 2024

1 Reviewer #1 (Remarks to the Author):

This paper reports an ordered sequence of models, from micro to meso to macro scale, and uses it to demonstrate a biological property of the brain, i.e., the state transitions occurring with deep anesthesia. The microscale, mesoscale, and macroscale models have been previously described in great detail in different papers, some also by the same authors. The Authors place an extended methodological description in most of the paragraphs of the result section, implying that this is indeed a paper on methods. Or better on a workflow, as they insist on the generality of their approach in several instances. Although the contribution to the bottom up modelling strategy is commendable, there are some points to consider.

We thank the Reviewer for this positive feedback. We provide below an answer to each of the points.

1) The propagation of the NMDA and GABA receptor properties from the spiking network to the mean field model to the virtual brain model is not explained. For example, the mean field model does not show any parameters directly related to NMDA or GABA receptors. The parameters of the mean field model that are closest to NMDA and GABA receptor currents, τ_e and τ_i , do not appear in any of the equations reported in the Methods. This explanation is crucial but is missing. An explanation here is needed, also for the sake of reproducibility.

We thank the Reviewer for the observation. We have edited the methods section to more clearly indicate where the properties of the NMDA and GABA receptors appear in the mean-field. The parameters τ_e and τ_i affect in particular the mean and variance of the synaptic conductances, now given in Eqs. 9-15

2) The paper is about the propagation of molecular level properties to higher scales, but the impact of pharmaceutical compounds is declared. The pharmacological issue here is a reflection of the simulation of changes in membrane receptor functions. There is no specific workflow to simulate drugs acting on the receptors though. This should be clarified.

We agree that this is not a “molecular model”, we model the effect of drugs on the kinetics and amplitude of the synaptic currents (or more generally, on membrane

conductances). However, the present approach would very well go in tandem with a molecular modeling approach where the effect of drugs on receptor kinetics is investigated - and the results of this molecular-level study could then be integrated in our approach. We have clarified this in the text. We have added the following sentence in the introduction: "The action of these anesthetics is incorporated here by modelling their effects on membrane receptors at single-cell level (a detailed simulation of the molecular action of drugs could also be incorporated, but is not adopted for the presentation of the framework in this paper).

3) The authors simulate state transition of cortical activity from wakefulness to sleep. The change is evident in the spiking network, in the mean field model, and in the virtual brain model. This is a nice *in silico* experiment. However, the discussion of results shown in Fig. 5 and 6 is very limited.

We appreciate your acknowledgment of our *in silico* experiments. We agree that the discussion of the results in Figures 5 and 6 requires further elaboration. As suggested, we expanded the discussion to provide deeper insights into the implications of the observed state transitions, emphasizing the results in Figures 5 and 6.

Also note that we have now expanded Figure 5 to include more conditions, not only Propofol (as before), but also Ketamine and NMREM sleep, so that Figs. 5 and 6 are more consistent. The model reproduces the experimental observation of a lack of effect of ketamine on FC/SC relation, unlike propofol and NREM sleep. The corresponding text was also expanded.

4) What about the firing rate of the E and I neurons ? By the way, in the mean field model two neurons represent more than 50 types of neurons of a cortical column that contains more than 30000 neurons as a whole. This should be explained. Any comparison between mean field model firing rates and those of the spiking network when anesthesia states are simulated ?

The comparison of the firing rates from the spiking network, mean field and whole-brain for the simulation of anesthesia are shown in Fig. 4 (second and third row). We show there how change in the parameters linked to the GABA and NMDA receptors generates the transition between the pattern of activity associated with the different brain-states (awake and anesthesia). We have now added a comment regarding different types of neurons and how this can be included in our formalism. In particular, populations representing specific neuronal types can be included in our formalism (implementations with up to 4 different neuronal types have been already implemented, see for example Lorenzi et al, PLOS Comp. Biol, 2023 or Tesler et al, biorxiv, 2024) and, in addition, heterogeneity on neuronal parameters (representing neurons with different properties within a certain distribution) can be incorporated in our model (see Di Volo and Destexhe, Sci. Reports, 2021).

5) Fig. 6 looks quite rich of information, but it is not described or commented beyond the figure legend. It must be described in order to make it understandable.

As suggested, we have revised the Results section to provide a more clear description and analysis of Figure 6. Additionally, we include a paragraph in the discussion section to ensure its rich information is clearly conveyed and properly interpreted.

2 Reviewer 2 (Remarks to the Author):

In this manuscript, the authors use a mean field based computational framework to evaluate pharmaceutical interventions on whole brain activity. They use a very interesting bottom up approach from single cell models to a whole brain network model so that the effect of pharmaceuticals can be introduced at the single cell microscale model parameters while observing the effects on a macroscale. This approach is innovative and significant for designing therapeutics specially in human subjects where we can only measure at the macroscopic level while the intervention works at the microscopic level. Using this approach, they show that anesthetics targeting GABA and NMDA receptors modelled as changes in the cellular membrane time constant can switch whole brain activity to generalized slow wave patterns as observed experimentally. This is an overall very well written paper with clear explanations.

Thank you for your positive feedback. We're grateful for your recognition of the innovation and significance of our framework. Your acknowledgment of the clarity in our explanations is especially encouraging. We answer to the different points below.

I have the following doubts regarding the methods that was not very clear to me and would appreciate some clarifications:

1) How are the single cell model parameters related to the mean field model parameter? In other words, how were the effects at the microscale incorporated in the macroscale model. For example, is the mean field time constant T related in some functional way to the τ 's that were manipulated to simulate effects of anesthetics. A description of this would be useful as this is the most salient feature of this paper.

Thank you for pointing to this lack of information, indeed we only referred to previous publications. We have now revised the Methods section to include a more detailed explanation on how micro-scale parameters associated with the anesthetics enter in the mean-field formalism. The parameters τ_e and τ_i enter directly in the calculation of the mean and variance of synaptic conductances, which is now given in Eqs. 9-15.

2) This is a forward model and produces output data that matches with experimental data. It would be useful for the reader to have a discussion on how one could use this framework to systematically estimate model parameters such as the τ s if one has the experimental data. For example, if we had the EEG/MEG recordings, can we use this data to find what model parameters are necessary to produce this observed brain recording. This would be in my opinion crucial for designing therapeutics and would take this approach one step closer to translation.

This is a very good remark, and actually we never attempted to do this kind of model inversion, to estimate parameters from large-scale data, as our focus was on the multiscale. However, one possibility would be to look at the frequency of network

oscillations (which can be recorded with any brain signal with sufficient resolution). This would require a separate study to integrate oscillations (for example gamma oscillations, beta oscillations etc) in the model and determine how the synaptic kinetics affects the oscillation. At this time, we would have a prediction of how these parameters could be estimated from measurements of the oscillation frequency (or other parameters such as synchrony). This is a beautiful possible follow-up of this approach and we have mentioned this in the manuscript. Thank you for the comment.

We have now modified the Discussion section in the following way: "To yield more precise models providing a quantitative comparison with experimental data, it is crucial to accurately reconstruct brain signals (such as EEG, MEG or MRI) from mean-field models, as proposed recently [Tesler et al, 2023], which will allow a more direct comparison with experiments. This also opens the possibility to estimate model parameters from these brain signals, and their variations due to the action of drugs of altered brain-states."

Reviewer 2 (Remarks on code availability):

I have browsed through the code and a readme file is present with instructions on which file to use. However, I have not run the code.

We have a readme file and additionally, we have included annotations that make the code even more readable and usable.

3 Reviewer 3 (Remarks to the Author):

In this paper, the authors have devised a framework that can build a whole brain model from a neuronal level, which can incorporate the dynamics of ion channels at the cellular level, spiking patterns at the microscale level and structural information from tractography data at the mesoscale level. This unique framework has four components – a spiking neuron model at a single neuron level, a network of spiking neuron models, a low dimensional mean field model incorporating the information of spiking neurons, and a whole brain model with the mean field model constrained by the structural connectivity. The current framework can be a valuable tool for deciphering the impact of the aberration of ion channels on neurological disorders or unconscious states at the macroscale or whole brain level. One of the applications of this model has been shown in the case of anaesthesia, where the pharmaceutical interval by Ketamine and Propofol is investigated both at the neural level (spiking pattern) and macroscale level (structure function) relationship and with perturbation.

The paper has exciting modelling aspects and has the potential to understand the impact of aberrations in neural dynamics at the whole brain level. The paper has several positive points, including the revised new methodology to make a more biologically plausible whole brain model from a high dimensional spiking network model to a low dimensional mass model (bottom up approach), the rigorous nature of the analysis, usage of the schematic diagram to discuss the underlying biological events (figure 1 and 2) and illustrating the application of the model to decipher the biological phenomena during the anaesthesia. The paper is suitable for publication, however the detailed revisions

is needed. However, some parts of the paper are unclear to the reviewer, which needs to be clarified.

Thank you for the positive and encouraging feedback. We greatly appreciate your recognition of the framework’s potential and its applications in understanding neural dynamics at the whole brain level. We carefully address the points of clarification in the revised manuscript to ensure the concepts are presented with greater clarity, as described below.

Firstly, the paper investigates the structure function relationship in reference and anaesthesia cases. However, it does not reveal how well the model can capture whole brain dynamics in terms of Functional connectivity, a traditional benchmark for the prediction power of the model. The authors can refer to the work of Luppi et al. (<https://www.nature.com/articles/s42003-022-03330-y>), where the model was tested with the Functional connectivity dynamics (FCD). A similar analysis can investigate the precision of the model.

Thanks for the suggestion. Our approach is different as the main focus is on multiscales. We obtain different biophysically inspired transitions from an asynchronous irregular regime to an UP-and-DOWN regime, at the level of firing rates. From these different regimes we simulate BOLD signals whose Functional connectivities are compared with the Structural connectivity (shown in Fig. 5), that show a well established result that the SC and FC’s are more correlated in states of low consciousness.

We have expanded Fig. 5 to now include more cases. This figure now shows the FC/SC relationship for three cases: Propofol (as before), NREM sleep and Ketamine. In each case, the FC/SC relation is compared with that of wakefulness. Interestingly, the model replicates the observation that ketamine produces minimal changes compared to wakefulness, unlike the changes observed for propofol anesthesia or NREM sleep. We have modified the figure to include these new analyses (based on published datasets) and simulations, as well as associated text.

Regarding functional connectivity dynamics (FCD), this is an excellent remark. We show here below an equivalent analysis as the one from Luppi et al. where we compare the KS-distance between the experimental propofol condition and the simulated FCD in awake and propofol conditions. This shows that our model also captures aspects of FCD. However, this analysis of experimental data sets and simulations (including optimization procedures as mentioned by the Reviewer), is a separate study (including unpublished data sets) that will be detailed in a forthcoming publication.

Secondly, the paper could better define the spiking network. It would be beneficial to include a description in the methods section regarding how the network was constructed and the rationale behind using 80% pyramidal (excitatory) neurons and 20% interneurons (inhibitory). A reference may suffice in this instance.

As suggested, we have extended the description of the spiking network and added the following citation for reference: Braitenberg, V. and Schüz, A. 2013. *Cortex: statistics and geometry of neuronal connectivity*. Springer-Verlag, Berlin.

Thirdly, the model investigates the underlying dynamics of the model by analyzing the excita-

Figure 1: Kolmogorov-Smirnov (KS) distances between the experimentally averaged functional connectivity dynamics (FCD) under the propofol condition and FCD derived from simulated data for the awake and propofol conditions. The KS distance is smaller for the simulated propofol condition compared to the simulated awake condition, supporting the validity of our simulation results.

tory firing pattern with the transfer function associated with it, and the transition state was shown with the loss of a stable self sustained state and the emergence of an up down state. However, it's difficult to follow this in section 2.6. "Emergence of UP and Down states." a basic description is needed to make the paper self sufficient. Some of the arguments from the earlier work by Di Volo et al.(2019) can be added to clarify the rationale behind referring to it as an 'Up and Down' state and the bistable behaviour of the model. As it is one of the central arguments of the work reflecting the impact of the parameters describing the interventions of propofol, it should be stated clearly.

We thank the Reviewer for this observation. We have now extended the description around the emergence of UP-and-DOWN states to make this aspect clearer.

Fourthly, the frequent references to the NREM sleep stage make the results section somewhat confusing. The comparison and analysis should either be discussed separately or mentioned in the discussion section. It is challenging to distinguish outcomes pertaining to anaesthesia cases from those related to sleep stages.

We have now modified the Results and Discussion sections all through to make it clearer. Since we now investigate the three conditions NREM sleep, propofol and ketamine, with respect to FC in the revised Fig. 5, we have clarified the discussion about these aspects as well.

Fifth, in the figures given in Fig. 6 A and B, it is impossible to follow the scattered plots for PCI (Perturbational Complexity Index) and make inferences from them. The authors can remove it by replacing it with a simple box plot with statistical analysis, which is enough to showcase the difference.

We thank the Reviewer for this comment. We point out that in Fig. 6.B we are adapting a figure with experimental data from a well known paper and we have thus maintained the original format. We have now improved the description in the caption of the figure to make it clearer. In addition, for Fig. 6.A (with simulated data) we include a box-plot with the statistical analysis to complement the scattered plot, as suggested by the reviewer. We hope this figure is now clear enough.

Sixth, In the Discussion, the paper lacks the proper comparative analysis. A brief discussion about the superiority of the model over the existing ones (i.e. Luppi et al., Deco et al. (2018), referenced in the paper but not discussed) can be described in terms of what can be inferred from the model, which is not possible from the other models. These arguments will make a strong case for this modelling approach.

We thank the Reviewer for this valuable suggestion to include a comparative analysis in the Discussion section. We clarify the unique strengths of our model relative to existing approaches, such as those based on macroscale observables like FCD by Luppi et al. and Deco et al. (2018). We now incorporate a brief discussion highlighting that our model adopts a bottom-up approach, offering a distinct advantage by effectively linking different scales of organization. Specifically, our model uniquely integrates molecular effects at the neuronal level, reproduces slow waves at the firing rate (mesoscale), and accounts for structure-function correlations at the macroscale. These capabilities enable our approach to propose mechanisms capable to reproduce different phenomena across scales, which are not fully addressed by the referenced models.

Seventh, the paper needs a parameter table to track the changes in parameters at a glance for different cases. Some of the parameters are not even available, like V_c in the equation in line number 684. It looks like it is a velocity parameter that incorporates the delay information. However, its importance has been shown in earlier mean field models like Castaldo et al. (2023) <https://doi.org/10.1016/j.neuroimage.2023.120236>.

We thank the Reviewer for this suggestion. We have now added a table (Table 2) with the parameters.

Lastly, the provided code is not annotated fully. It's difficult to follow it seamlessly and produce the results given in the paper. Authors can add a parameter file or clearly mention the parameters involved in the paper to produce the results.

As suggested, we have now extended the annotations in the code. All the parameters used in the code can be found in the file:
`./TVB/tvb_model_reference/simulation_file/parameter/parameter_M_Berlin_new.py.`

The values of adaptation strength (b_e), synaptic decay (τ_e, τ_i) vary in our simulations across the different states and can be found on the newly added parameters table (Table 2) in the manuscript.

Minor revisions that must be addressed.

1. In the summary of Figure 4, the authors mention UD. What is it?

It refers to Up and Down state. We have added the definition.

2. In line 408, the authors talked about correlating SC with FC. The term “Reference SC” is used. Please clarify it. Is it a mean SC of all the individuals or an individual SC? Is the model a personalized model meaning a separate SC used to compute the FC of each person?

We thank the reviewer for highlighting the need to clarify the term “Reference SC.” In the revised manuscript, we clarified this to ensure precision and avoid ambiguity. The term “Reference SC” refers to the SC used both to generate data with our model and to compute the correlations between SC and the corresponding individual experimental FCs or the simulated FCs, which are obtained using different random seeds for the noise term in our equations. The reference SC used in this work comes from connectivity data from a cohort of unrelated healthy adults from the Human Connectome Project (HCP) (n=207; 83 males, in a range=22-36 years). The SC represent the group average normative structural connectome. In this paper the topic of personalized models was not treated, but the formalism and workflow are certainly compatible with the use of personalised SC and models. We mention this in the discussion.

3. The chronology of numbering for the equations is not followed; some are numbered, and some are not. Please enumerate them.

We have fixed this in the revised version of the manuscript.

4. In line 743, the phrase “significant vectors...” sounds vague. Rephrase it.

We have now added a more detailed explanation of the PCI calculation to improve clarity: “The firing rates of the excitatory populations were first normalized using the mean and standard deviation of the pre-stimulus interval. To assess statistical significance, a bootstrap procedure was employed to generate a null distribution of surrogate values. This was done by repeatedly shuffling the pre-stimulus signals. A significance threshold was then determined from the null distribution. Finally, the normalized post-stimulus firing rates were compared to this threshold, resulting in a binary map of significant vectors $s(t)$ for each trial.”